# THE CURSE OF DIVERSITY IN ENSEMBLE-BASED EXPLORATION

**Zhixuan Lin,** *Pierluca D'Oro, Evgenii Nikishin & Aaron Courville
Mila - Quebec AI Institute, Université de Montréal

## ABSTRACT

We uncover a surprising phenomenon in deep reinforcement learning: training a diverse ensemble of data-sharing agents – a well-established exploration strategy – can significantly impair the performance of the individual ensemble members when compared to standard single-agent training. Through careful analysis, we attribute the degradation in performance to the low proportion of self-generated data in the shared training data for each ensemble member, as well as the inefficiency of the individual ensemble members to learn from such highly off-policy data. We thus name this phenomenon *the curse of diversity*. We find that several intuitive solutions – such as a larger replay buffer or a smaller ensemble size – either fail to consistently mitigate the performance loss or undermine the advantages of ensembling. Finally, we demonstrate the potential of representation learning to counteract the curse of diversity with a novel method named Cross-Ensemble Representation Learning (CERL) in both discrete and continuous control domains. Our work offers valuable insights into an unexpected pitfall in ensemble-based exploration and raises important caveats for future applications of similar approaches.

## 1 INTRODUCTION

Ensemble-based exploration, i.e. training a diverse ensemble of data-sharing agents, underlies many successful deep reinforcement learning (deep RL) methods (Osband et al., 2016; 2018; Liu et al., 2020; Schmitt et al., 2020; Peng et al., 2020; Hong et al., 2020; Januszewski et al., 2021). The potential benefits of a diverse ensemble are twofold. At training time, it enables concurrent exploration with multiple distinct policies without the need for additional samples. At test time, the learned policies can be aggregated into a robust ensemble policy, via aggregation methods such as majority voting (Osband et al., 2016) or averaging (Januszewski et al., 2021).

Despite the generally positive perception of ensemble-based exploration, we argue that this approach has a potentially negative aspect that has been long overlooked. As shown in Figure 1, for each member in a data-sharing ensemble, only a small proportion of its training data comes from its own interaction with the environment. The majority of its training data is generated by other members of the ensemble, whose policies might be distinct from its own policy. This type of off-policy learning has been shown to be highly challenging in previous work (Ostrovski et al., 2021). We thus hypothesize that similar learning difficulties can also occur in ensemble-based exploration.

We verify our hypothesis in the Arcade Learning Environment (Bellemare et al., 2012) with the Bootstrapped DQN (Osband et al., 2016) algorithm and the Gym MuJoCo benchmark (Towers et al., 2023) with an ensemble SAC (Haarnoja et al., 2018a) algorithm. We show that, in many environments, the individual members of a data-sharing ensemble significantly underperform their single-agent counterparts. Moreover, while aggregating the policies of all ensemble members via voting or averaging sometimes compensates for the degradation in individual members' performance, it is not always the case. These results suggest that ensemble-based exploration has a hidden negative effect that might weaken or even completely eliminate its advantages. We perform a series of experiments to confirm the connection between the observed performance degradation and the off-policy learning challenge posed by a diverse ensemble. We thus name this phenomenon *the curse of diversity*.

---

*Correspondence to `zxlin.cs@gmail.com`.

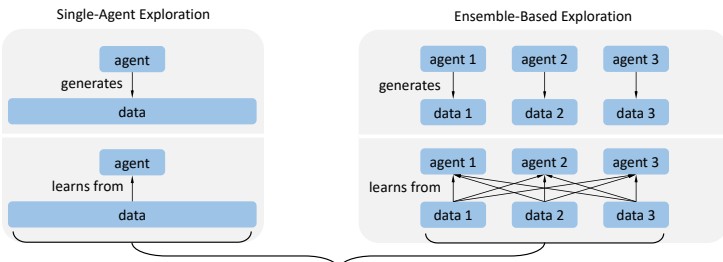

Figure 1: Comparison between standard single-agent exploration and ensemble-based exploration. In single-agent training, one agent generates and learns from all the data. In ensemble-based exploration with $N$ ensemble members, each agent generates $1/N$ of the data but learns from all the data.

We show that several intuitive solutions – such as a larger replay buffer or a smaller ensemble size – either fail to consistently mitigate the performance loss or undermine the advantages of ensembling. Inspired by previous work's finding that network representations play a crucial role in related settings (Ostrovski et al., 2021; Kumar et al., 2021), we investigate whether representation learning can mitigate the curse of diversity. Specifically, we propose a novel method named Cross-Ensemble Representation Learning (CERL) in which individual ensemble members learn each other's value function as an auxiliary task. Our results show that CERL mitigates the curse of diversity in both Atari and MuJoCo environments and outperforms the single-agent and ensemble-based baselines when combined with policy aggregation.

We summarize our contributions as follows:

1. We expose *the curse of diversity* phenomenon in ensemble-based exploration: individual members in a data-sharing ensemble can vastly underperform their single-agent counterparts.

2. We pinpoint the off-policy learning challenges posed by a diverse ensemble as the main cause of the curse of diversity and provide extensive analysis.

3. We show the potential of representation learning to mitigate the curse of diversity with a novel method named Cross-Ensemble Representation Learning (CERL) in both discrete and continuous control domains.

## 2 PRELIMINARIES

We outline some important specifications that we use throughout this work.

**Ensemble-based exploration strategy**    We focus our discussion on a simple ensemble-based exploration strategy that underlies many previous works (Osband et al., 2016; 2018; Liu et al., 2020; Schmitt et al., 2020; Peng et al., 2020; Hong et al., 2020; Januszewski et al., 2021), depicted in Figure 1. The defining characteristics of this strategy are as follows:

1. *Temporally coherent exploration:* At training time, within each episode, only the policy of one ensemble member is used for selecting the actions.

2. *Relative independence between ensemble members:* Each ensemble member has its own policy (may be implicit), value function, and target value function. Most importantly, the regression target in Temporal Difference (TD) updates should be computed separately for each ensemble member with their own target network.

3. *Off-policy RL algorithms with a shared replay buffer:* Different ensemble members share their collected data via a central replay buffer (Lin, 1992). To allow data sharing, the underlying RL algorithm should be off-policy in nature.

**Environments and algorithms**    We use 55 Atari games from the Arcade Learning Environment and 4 Gym MuJoCo tasks for our analysis. We train for 200M frames for Atari games and 1M steps for MuJoCo tasks. For aggregate results over 55 Atari games, we use the interquartile mean (IQM) of human-normalized scores (HNS) as recommended by Agarwal et al. (2021). In some results where

HNS is less appropriate, we exploit another score normalization scheme where Double DQN and a random agent have a normalized score of $1$ and $0$ respectively. This will be referred to as Double DQN normalized scores. More experimental details can be found in Appendix B.

For our analysis, we use Bootstrapped DQN (Osband et al., 2016) with Double DQN updates (Hasselt et al., 2015) for Atari and an ensemble version of the SAC (Haarnoja et al., 2018b) algorithm (referred to as *Ensemble SAC* in the following) for MuJoCo tasks. Ensemble SAC follows the same recipe as Bootstrapped DQN except with SAC as the base algorithm. We provide pseudocode for these algorithms in Appendix A. Correspondingly, for the single-agent baselines, we use Double DQN and SAC. For analysis purposes, for continuous control, we use a replay buffer of size 200k by default (as opposed to the usual 1M) since we find the curse of diversity is more evident with smaller replay buffers in these tasks. Also, to avoid the confounding factor of representation learning, we do not share the networks across the ensemble in Bootstrapped DQN by default. These factors will be analyzed in Section 3.3. Following Osband et al. (2016)'s setup for Atari, we do not use data bootstrapping in our analysis. Throughout this work, we use $L$ to denote the number of shared layers across the ensemble members and $N$ to denote the ensemble size. Unless otherwise specified, we use $L = 0$ and $N = 10$. For each ensemble algorithm $X$ – where $X$ can be either Bootstrapped DQN or Ensemble SAC – we consider two different evaluation methods:

1. *X (aggregated)* or *X (agg.)*: we aggregate the policies of all ensemble members during testing. For discrete action tasks, we use majority voting (Osband et al., 2016). For continuous control, we average the actions of all policies as in Januszewski et al. (2021).

2. *X (individual)* or *X (indiv.)*: for each evaluation episode, we randomly sample one ensemble member for acting. This interaction protocol is exactly the same as the one used during training and aims to measure the performance of the individual ensemble members.

We emphasize that these two methods *only differ at test-time*. We only train $X$ once and then obtain the results of *X (agg.)* and *X (indiv.)* using the above interaction protocols during evaluation. *Policy aggregation is never used during training*. Also, the performance of *X (agg.)* is often what we eventually care about. More implementation details can be found in Appendix C.

## 3 THE CURSE OF DIVERSITY

The central finding of this paper is as follows:

> Individual ensemble members in a data-sharing ensemble can suffer from severe performance degradation relative to standard single-agent training due to: (1) the low proportion of self-generated data in the shared training data for each ensemble member; and (2) the inefficiency of the individual ensemble members to learn from such highly off-policy data.

We name the performance degradation in individual ensemble members due to challenging off-policy learning posed by a diverse ensemble *the curse of diversity*. In the following, we demonstrate this phenomenon, verify its cause, and provide extensive analysis.

### 3.1 THE NEGATIVE EFFECT OF ENSEMBLE-BASED EXPLORATION

We show the curse of diversity phenomenon with $55$ Atari games and $4$ MuJoCo tasks in Figure 2. These results show a clear underperformance of the individual ensemble members (e.g., Bootstrapped DQN (indiv.)) relative to their single agent counterparts (e.g., Double DQN). Note that even though they are trained on different data distributions and thus are expected to behave differently, the agents in these two cases *have access to the same amount of data and have the same network capacity*. The underperformance happens in the majority of Atari games and $3$ out of the $4$ MuJoCo tasks, suggesting that this is a universal phenomenon. Surprisingly, simply aggregating the learned policies *at test-time* provides a huge performance boost in many environments, and in many cases fully compensates for the performance loss in the individual policies (e.g., Walker2d). This partially explains why previous works – which often only report the performance of the aggregated policies (Osband et al., 2016; Chiang et al., 2020; Agarwal et al., 2020; Meng et al., 2022) – fail to notice this phenomenon.

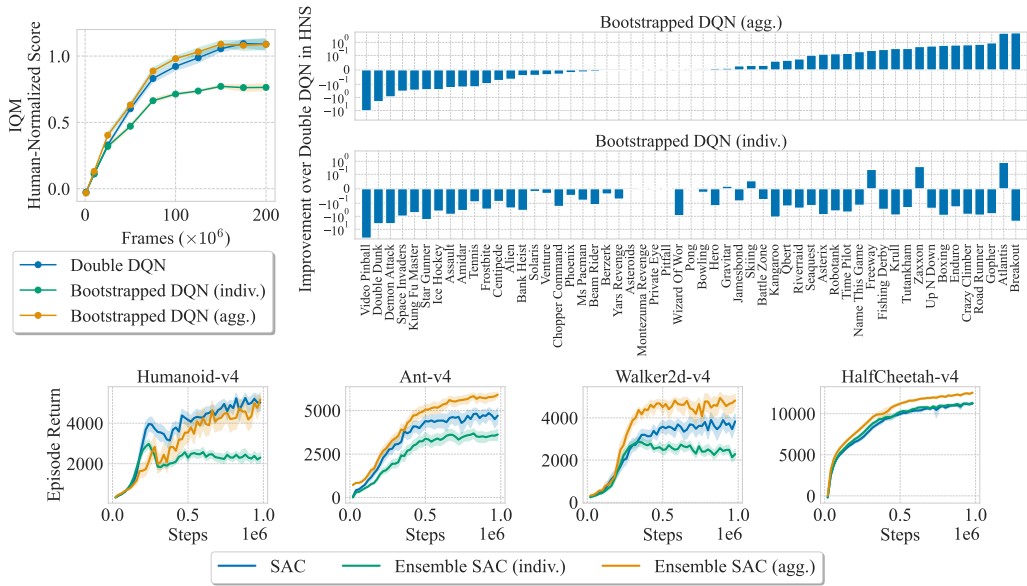

Figure 2: (**top-left**) Comparison between Double DQN, Bootstrapped DQN (agg.), and Bootstrapped DQN (indiv.) in 55 Atari games. Shaded areas show 95% bootstrapped CIs over 5 seeds. (**top-right**) Per-game performance improvement of Bootstrapped DQN (indiv.) and Bootstrapped DQN (agg.) over Double DQN, measured as the difference in HNS. All methods use a replay buffer size of 1M. (**bottom**) Comparison between SAC, Ensemble SAC (indiv.) and Ensemble SAC (agg.) in 4 MuJoCo tasks with a replay buffer size of 200k. Shaded areas show 95% bootstrapped CIs over 30 seeds. All ensemble methods in this figure use $N = 10$ and $L = 0$.

The significance of these results is threefold. First, it challenges some previous work's claims that the improved performance of approaches such as Bootstrapped DQN in some tasks is mainly due to better exploration. As shown in Figure 2, in most games where Bootstrapped DQN (agg.) performs better than Double DQN, Bootstrapped DQN (indiv.) significantly underperforms Double DQN. This means most benefits of Bootstrapped DQN in these games come from majority voting, which is largely orthogonal to exploration. However, this does not imply that better exploration – and hence wider state-action space coverage – brought by a diverse ensemble is not beneficial, as its effects might be overshadowed by the curse of diversity and thus not visible. Second, it raises important caveats for future applications of ensemble-based exploration, especially in certain scenarios such as hyperparameter sweep with ensembles (Schmitt et al., 2020; Liu et al., 2020) where we mainly care about individual agents' performance. Finally, it presents an opportunity for better ensemble algorithms that mitigate the curse of diversity *while preserving the advantages of using an ensemble*. To this end, we perform an analysis to better understand the cause of the performance degradation.

### 3.2 UNDERSTANDING ENSEMBLE PERFORMANCE DEGRADATION

We hypothesize that the observed performance degradation is due to (1) the low proportion of self-generated data in the shared training data for each ensemble member, and (2) the inefficiency of the individual ensemble members to learn from such highly off-policy data. Inspired by the tandem RL setup in Ostrovski et al. (2021), we design a simple "$p$%-tandem" setup to verify this hypothesis. Similar to the original tandem RL setup, we train a pair of active and passive agents *sharing the replay buffer and training batches*. For each training episode, with probability $1 - p\%$ we use the active agent for acting; otherwise, we use the passive agent for acting. In other words, the active agent generates $1 - p\%$ of the data and the passive agent generates $p\%$ of the data. Note that this is different from the "$p$% self-generated data" experiment in Ostrovski et al. (2021) as they use two separate buffers for the two agents. In contrast, in our $p$%-tandem setup, the two agents share the replay buffer and the training batches, and thus *any performance gap between the active and passive*

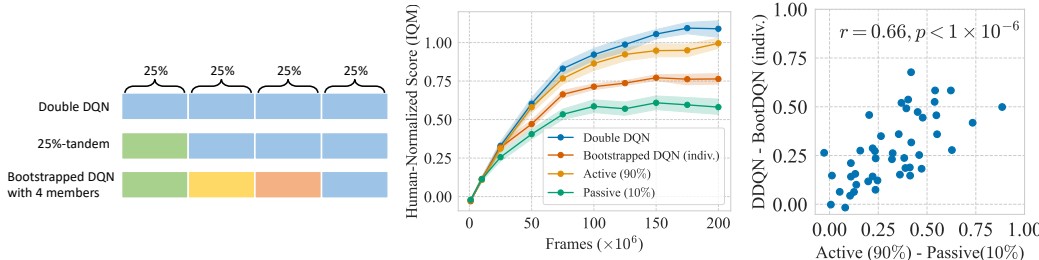

Figure 3: (**left**) Different algorithms as variants of the same ensemble algorithm, using $N = 4$ as an example. Each block represented $25\%$ of the generated data. Data blocks of the same colors are generated by identical agents. (**middle**) Comparison between Double DQN, Bootstrapped DQN (indiv.) with $N = 10$ and $L = 0$, and the active and passive agents in the $10\%$-tandem setup. All methods use a replay buffer of size 1M. Shaded areas show $95\%$ bootstrapped CIs. Results are aggregated over 5 seeds and 55 games. (**right**) Correlation between (1) the performance gap between the active and passive agents and (2) the performance gap between Double DQN and Bootstrapped DQN (indiv.) in different games. Each point corresponds to a game. We use Double DQN normalized scores instead of HNS since the scale of the latter can vary a lot across games. Eight games where Double DQN's performance is close to random (HNS $< 0.05$), and one game whose data point lies on the negative half of the $y$-axis in the plot are omitted since they trivially satisfy our hypothesis.

*agents can only be due to the difference in the proportions of the two agents' self-generated data and the inefficiency of the passive agent to learn from the shared data.*

To understand why $p\%$-tandem may support our hypothesis, it is useful to view Double DQN, $p\%$-tandem with two Double DQN agents, and Bootstrapped DQN as variants of the same ensemble algorithm with $N$ members, where each member generates $1/N$ of the data. Taking $N = 4$ and $p\% = \frac{1}{N} = 25\%$ as an example, we have the following *exact* equivalence (shown in Figure 3 (left)):

ALGO 1 Double DQN $\Leftrightarrow$ 4 ensemble members, but all of them are identical;

ALGO 2 $25\%$-tandem $\Leftrightarrow$ 4 ensemble members, but the last 3 of them are identical;

ALGO 3 Bootstrapped DQN $\Leftrightarrow$ 4 ensemble members, and all of them are different.

Our core reasoning is as follows. If the first member in ALGO 2 (i.e. the passive agent) suffers from severe inefficiency to learn from the shared data (i.e., it significantly underperforms the active agent), it should also suffer from the same learning inefficiency if we replace the other 3 identical members (i.e., the active agent) with 3 different members, which is just ALGO 3/Bootstrapped DQN. Further, if the performance gap between the active and passive agents is comparable with or larger than the performance gap between Double DQN and Bootstrapped DQN (indiv.), then the observed learning inefficiency is *sufficient* to cause the performance degradation we see in Bootstrapped DQN (indiv.).

**Verifying the hypothesis** In Figure 3 (middle) we show the performance of Double DQN, Bootstrapped DQN (indiv.) with $N = 10$, and the active and the passive agents in the $10\%$-tandem setup. As expected, we see that the passive agent significantly underperforms the active agent even though they share the training batches, indicating the inefficiency of the passive agent to learn from the shared data. Also, the performance gap between the active and passive agents is comparable to the performance gap between Double DQN and Bootstrapped DQN (indiv.). A similar analysis for MuJoCo tasks is presented in Appendix D.1 and shows similar patterns. In Figure 3 (right), we show a clear correlation between (1) the performance gap between the active and the passive agents and (2) the performance gap between Double DQN and Bootstrapped DQN (indiv.) in different games. These results offer strong evidence of a connection between the off-policy learning challenges in ensemble-based exploration and the observed performance degradation.

**Remark on data coverage** We comment that another important aspect of having a diverse ensemble is *wider state-action space coverage* in the data. Even though the degree of "off-policy-ness" and the state-action space coverage are often correlated in practice, they are different: state-action space coverage is *purely a property of the data distribution*, while "off-policy-ness" involves *both the data distribution and the policies*. Our $p\%$-tandem experiment disentangles these two aspects by

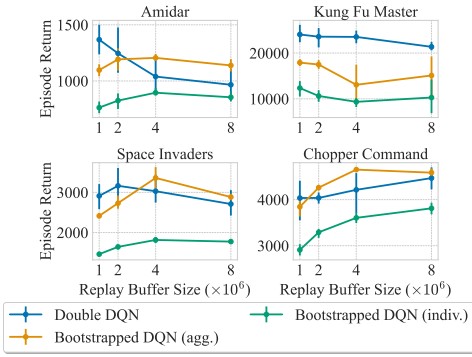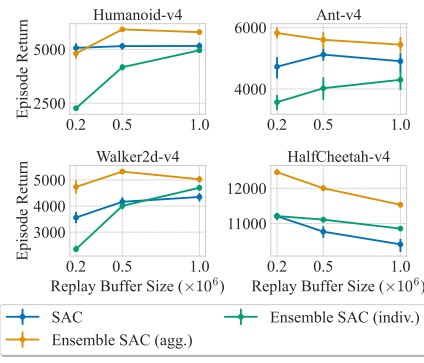

Figure 4: (**left**) The effects of replay buffer size in 4 Atari games. Error bars show 95% bootstrapped CIs over 5 seeds. (**right**) The effects of replay buffer size in 4 MuJoCo tasks. Error bars show 95% bootstrapped CIs over 30 seeds. We $N = 10$ and $L = 0$ for Bootstrapped DQN.

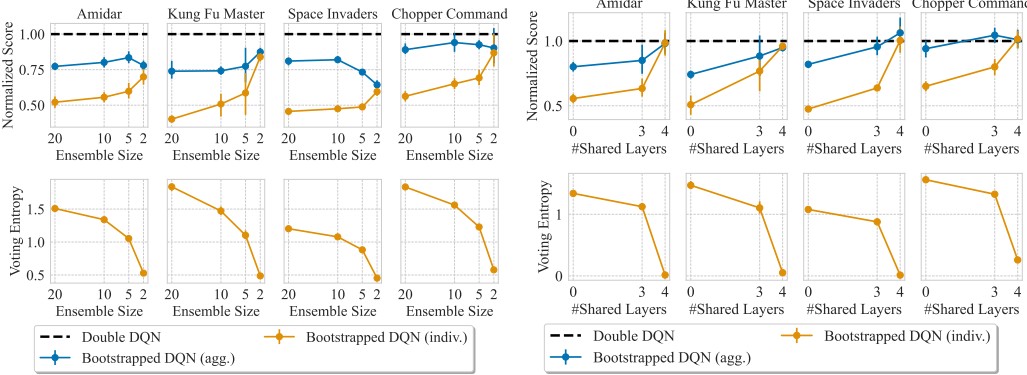

Figure 5: (**left**) The effects of adjusting the ensemble size. We use $L = 0$ for Bootstrapped DQN. (**right**) The effects of varying the number of shared layers. We use $N = 10$ for Bootstrapped DQN. The top rows show Double DQN normalized scores. The bottom rows show the entropy of the normalized vote distributions. Error bars show 95% bootstrapped CIs over 5 seeds. All methods use a replay buffer of size 1M.

having the active and passive agents trained on the same data (thus the same coverage) but experience different degrees of "off-policy-ness". Therefore, the active/passive gap implies that off-policy-ness is detrimental, but *it does not indicate whether wider state-action space coverage is beneficial or harmful*. On the other hand, the fact that (1) Bootstrapped DQN (indiv.) likely suffers from greater "off-policy-ness" than the passive agent due to the presence of more policies and that (2) Bootstrapped DQN (indiv.) outperforms the passive agent in Figure 3 indicates that the wider state-action space coverage of the data generated by Bootstrapped DQN (indiv.) is likely highly beneficial. We leave a rigorous analysis of the effect of state-action space coverage for future work.

### 3.3 MITIGATING THE CURSE OF DIVERSITY: INITIAL ATTEMPTS

Having established the main cause of observed performance degradation, we examine whether several intuitive solutions can mitigate it in this section. More analysis can be found in Appendix D.3.

**Larger replay buffers**  A larger replay buffer provides better state-action coverage and may mitigate issues due to insufficient self-generated data such as erroneous value extrapolation (Fujimoto et al., 2018a; Kumar et al., 2019; Ostrovski et al., 2021). In Figure 4, we probe the effects of replay buffer capacity in 4 Atari games and 4 MuJoCo tasks. For MuJoCo tasks, a larger replay buffer mitigates the curse of diversity (i.e., reduces the performance gap between SAC and Ensemble SAC (indiv.)) in Humanoid and Walker, though it still remains in Ant. However, in Atari games, the performance gap between Double DQN and Bootstrapped DQN (indiv.) largely remains with larger replay buffers,

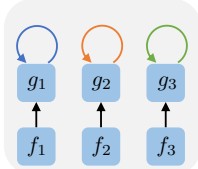 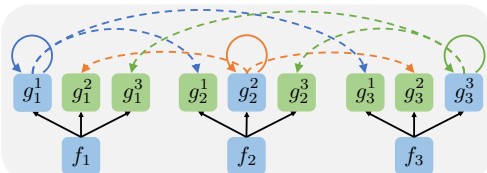

Figure 6: (**left**) The Q-value networks of a standard ensemble without CERL. (**right**) The Q-value networks of an ensemble with CERL. $f$ and $g$ represent neural networks. An arrow from $X$ to $Y$, where $X$ and $Y$ are either $g_i$ or $g_i^j$, indicates $X$ will be used as the target network of $Y$ when performing TD updates. Auxiliary heads are in green. Dashed lines indicate the corresponding loss terms are CERL auxiliary losses. Arrows with the same color originate from the same main head.

except for Amidar where the performance of Double DQN itself is severely impaired. The difference in results in the two domains may be due to the different numbers of samples required (1M versus 200M). It is possible that an even larger replay buffer may mitigate the curse of diversity in Atari, but this is extremely expensive and infeasible for us to test. Overall, these results suggest that though increasing the replay buffer capacity (within a reasonable memory budget) can mitigate the curse of diversity in some environments, it is not a consistent remedy.

**Reducing diversity** The most intuitive way to mitigate the curse of diversity is, naturally, by reducing diversity. We test two techniques to do so: (1) reducing the ensemble size and (2) increasing the number of shared layers across the ensemble. In Figure 5, we show the impact of these two techniques on performance and ensemble diversity in 4 Atari games. To quantify diversity, we measure the entropy of the distribution of votes among the ensemble members. As expected, both techniques reduce diversity and improve the performance of Bootstrapped DQN (indiv.). However, as the diversity of the ensemble decreases the advantages of policy aggregation also reduce, which can be seen from the tapering performance gap between Bootstrapped DQN (indiv.) and Bootstrapped DQN (agg.). These results show that even though reducing diversity can mitigate the curse of diversity, it also compromises the advantages of policy aggregation. Ideally, we would want a solution that alleviates the curse of diversity *while preserving the advantages of using ensembles*. In the next section, we show that representation learning offers a promising solution.

## 4 MITIGATING THE CURSE OF DIVERSITY WITH REPRESENTATION LEARNING

Our method is motivated by a simple hypothesis. We conjecture that the reason why sharing network layers mitigates the curse of diversity is twofold. First, as mentioned above, sharing layers reduces ensemble diversity, thus making the generated data "less off-policy" to the individual members. Second, the shared network *simultaneously learns the value functions of multiple ensemble members*, leading to improved representations. This may lead to more efficient off-policy learning by allowing the Q-value networks to better generalize to state-action pairs that have high probability under the current policy but are under-represented in the data. As we have shown, the diversity reduction aspect has the undesirable side effect of reducing the benefits of policy aggregation. We thus propose *Cross-Ensemble Representation Learning* (CERL), a novel method that benefits from the similar representation learning effect of network sharing without actually needing to share the networks, thus preserving the diversity of the ensemble.

**Method** CERL, shown in Figure 6 (right), is an auxiliary task that can be applied to most ensemble-based exploration methods that follow the recipe we outline in Section 2. For ease of exposition, we use Ensemble SAC as an example. Extension to other methods is trivial. In CERL, for each Q-value network, we conceptually split it into an encoder $f$ and a head $g$. Our goal in CERL is to force the Q-value network encoder of *each* ensemble member to learn the value functions of *all* the ensemble members, similar to what happens when using explicit network sharing. To this end, *each* ensemble member $i$ has $N$ Q-value heads $\{Q_i^j(s,a) = [g_i^j(f_i(s,a))]\}_{j=1}^N$. For ensemble member $i$, $Q_i^i(s,a)$ is the "main head" that *defines* member $i$'s value function estimation. Each ensemble member $i$ still has *only one* policy $\pi_i(a|s)$, and the main head $Q_i^i(s,a)$ is the only head used to update the policy. A head $Q_i^j(s,a)$ of ensemble member $i$ where $j \neq i$ is used to learn the value function of ensemble member $j$ as an auxiliary task. These heads are referred to as "auxiliary heads" because their sole purpose is

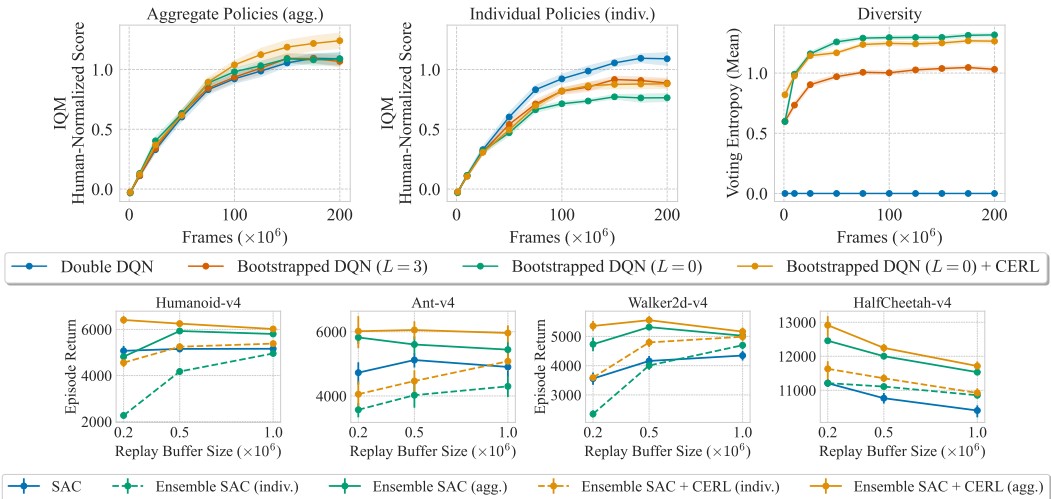

Figure 7: (**top**) Comparison between Double DQN, Bootstrapped DQN, and CERL in Atari. Results are aggregated over 55 games and 5 seeds. We show the performance of the agg. and indiv. versions of each ensemble algorithm in the top left and top middle plots respectively. Shaded areas show 95% bootstrapped CIs. All methods use a replay buffer of size 1M. (**bottom**) Comparison between SAC, Ensemble SAC, and CERL across different replay buffer sizes in MuJoCo tasks. Error bars show 95% bootstrapped CIs over 30 seeds. All ensemble methods in this figure uses $N = 10$.

to provide better representations for the main heads. Specifically, given a transition $(s, a, r, s')$, we perform the following TD update for all $N \times N$ heads in parallel as follows:

$$Q_i^j(s, a) \leftarrow r + \gamma \bar{Q}_j^j(s', a_j'), \quad \text{for } i = 1, \dots, N, \quad \text{for } j = 1, \dots, N \qquad (1)$$

where $\bar{Q}_j^j$ is the target network for $Q_j^j$ and $a_j' \sim \pi_j(\cdot|s')$. As usual, the update rule is implemented as a one-step stochastic gradient descent.

Besides being conceptually simple, CERL is easy to implement. In our experiments, we find it sufficient to duplicate the last linear layers of the networks as the auxiliary heads, which can be implemented by increasing the networks' final output dimensions. For the same reason, CERL is computationally efficient. For example, for the Nature DQN network (Mnih et al., 2015) used in this work, applying CERL to Bootstrapped DQN with $N = 10$ and $L = 0$ increases the number of parameters by no more than 5%, and the increase in wall clock time is barely noticeable. We provide pseudocode for CERL with Ensemble SAC and Bootstrapped DQN in Appendix A.

**Experiments** We focus on two questions: (1) Can CERL mitigate the curse of diversity, i.e. the performance gap between individual ensemble members and their single-agent counterparts? (2) Do the improvements in individual ensemble members translate into a better aggregate policy? To answer these questions, we test CERL on Bootstrapped DQN in 55 Atari games (Figure 7 (top)) and on Ensemble SAC in 4 MuJoCo tasks (Figure 7 (bottom)). We compare with Bootstrapped DQN and Ensemble SAC without CERL as well as the single-agent Double DQN and SAC. To show the advantage of CERL over explicit network sharing, we also include Bootstrapped DQN with network sharing and report ensemble diversity as we did in Section 3.3. We use $L$ to denote the number of shared layers across the ensemble. As the curse of diversity is sensitive to replay buffer size in MuJoCo tasks, we show results with different replay buffer sizes for these tasks. Additional results, including ensemble size ablations and an alternative design of CERL, can be found in Appendix D.4.

As shown in these results, CERL consistently mitigates the curse of diversity across the tested environments. For example, applying CERL to Ensemble SAC in Humanoid with a 0.2M-sized replay buffer reduces the performance gap between SAC and Ensemble SAC (indiv.) from roughly 3000 to around 500. More importantly, the improvements in individual policies *do* translate into improvements in aggregate policies, which enables CERL to achieve the best performance with policy aggregation in both domains. In contrast, even though Bootstrapped DQN ($L = 3$) also performs better than Bootstrapped DQN ($L = 0$) when comparing individual policies, it does not provide any

gain over Bootstrapped DQN ($L = 0$) when comparing aggregate policies, likely due to significantly lower ensemble diversity than Bootstrapped DQN ($L = 0$) as shown in Figure 7 (top right).

## 5 RELATED WORK

**Ensemble-based exploration** The idea of training an ensemble of data-sharing agents that concurrently explore has been employed in many deep RL algorithms (Osband et al., 2016; 2018; Liu et al., 2020; Schmitt et al., 2020; Peng et al., 2020; Hong et al., 2020; Januszewski et al., 2021). Most of these works focus on algorithmic design and the discussion of the potential negative effects of the increased "off-policy-ness" compared to single-agent training has largely been missing. To the best of our knowledge, the only work that explicitly discusses the potential difficulties of learning from the off-policy data generated by other ensemble members is Schmitt et al. (2020). However, since Schmitt et al. (2020) does not maintain explicit Q-values and relies on V-trace (Espeholt et al., 2018) for off-policy correction, algorithmic changes that allow stable off-policy learning are a *requirement* in their work. In contrast, we show that even for Q-learning-based methods that do not require explicit off-policy correction, ensemble-based exploration can still lead to performance degradation. There also exist methods that use multiple ensemble members on a *per-step* basis (Chen et al., 2018; Lee et al., 2021; Ishfaq et al., 2021; Li et al., 2023) as opposed to one ensemble member for each episode. The discussion of these methods is more subtle and is left for future work. Sun et al. (2022) uses an ensemble of Q-value networks to trade off exploration and exploitation. However, their method only uses one policy so our discussion does not apply to it.

**Ensemble RL methods for other purposes** Ensemble methods have also been employed in RL for purposes other than exploration, for example to produce robust value estimations (Anschel et al., 2016; Lan et al., 2020; Agarwal et al., 2020; Peer et al., 2021; Chen et al., 2021; An et al., 2021; Wu et al., 2021; Liang et al., 2022) or model predictions (Chua et al., 2018; Kurutach et al., 2018). Despite the use of ensembles, most of these methods are still single-agent in nature (i.e., there is only one policy interacting with the environment). Thus, our discussion does not apply to these methods.

**Mutual distillation** Mutual/collaborative learning in supervised learning (Zhang et al., 2017; Anil et al., 2018; Guo et al., 2020; Wu & Gong, 2020) aims to train a cohort of networks and share knowledge between them via mutual distillation. Similar ideas have also been employed in RL (Czarnecki et al., 2018; Xue et al., 2020; Zhao & Hospedales, 2020; Reid & Mukhopadhyay, 2021). CERL is distinct from these works in that it does not try to distill different ensemble members' predictions (i.e., the value functions) into each other. Instead, CERL is only an auxiliary task, and different ensemble members in CERL only affect each other's *representations* via auxiliary losses.

**Auxiliary tasks in RL** Facilitating representation learning with auxiliary tasks has been shown to be effective in RL (Jaderberg et al., 2016; Mirowski et al., 2016; Fedus et al., 2019; Kartal et al., 2019; Dabney et al., 2020; Schwarzer et al., 2020). In the context of multi-agent RL, He & Boyd-Graber (2016), Hong et al. (2017); Hernandez-Leal et al. (2019) and Hernandez et al. (2022) model the policies of *external* agents as an auxiliary task, and Barde et al. (2019) promotes coordination between several trainable agents by maximizing their mutual action predictability. Besides the clear differences in the problem domains and motivations, these multi-agent works only predict the *actions* of other agents, which typically contain less information than the $Q$-function used in CERL's auxiliary task.

## 6 DISCUSSION AND CONCLUSION

In line with recent efforts to advance the understanding of deep RL through purposeful experiments (Ostrovski et al., 2021; Schaul et al., 2022; Nikishin et al., 2022; Sokar et al., 2023), our work builds on extensive, carefully designed empirical analyses. It offers valuable insights into a previously overlooked pitfall of the well-established approach of ensemble-based exploration and presents opportunities for future work. As with most empirical works, an important avenue for future research lies in developing a theoretical understanding of the phenomenon we reveal in this work.

A limitation of CERL is its reliance on separate networks for high ensemble diversity, which may become infeasible with very large networks. A simple improvement to CERL is thus to combine CERL with network sharing and encourage diversity with other mechanisms, such as randomized prior functions (Osband et al., 2018) and explicit diversity regularization (Peng et al., 2020).

## REPRODUCIBILITY STATEMENT

Detailed pseudocode of the ensemble algorithms used in this work is provided in Appendix A. Experimental and implementation details are given in Appendix B and Appendix C respectively. The source code is available at the following repositories:

- Atari: `https://github.com/zhixuan-lin/ensemble-rl-discrete`
- MuJoCo: `https://github.com/zhixuan-lin/ensemble-rl-continuous`

## ACKNOWLEDGMENTS

This research was enabled in part by support and compute resources provided by Mila (`mila.quebec`), Calcul Québec (`www.calculquebec.ca`), and the Digital Research Alliance of Canada (`alliancecan.ca`). We thank Sony for their financial support of Zhixuan Lin throughout this work.

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

# Appendix

## Table of Contents

# A    ALGORITHMS

In this section, we provide the pseudocode for the ensemble algorithms used in this work. For simplicity, we only describe the case of batch size $B = 1$. For $B > 1$ we simply average the loss across the batch. Note we use $\theta$ and $\phi$ to denote the parameters of the *entire ensemble*. In other words, $Q_i(s, a; \theta)$ only depends on a subset of $\theta$. All ensemble algorithms do not use data bootstrapping, as Osband et al. (2016) finds it barely affects performance in complex domains. All ensemble gradient updates are performed in parallel in practice, though in the pseudocode they are written as for loops.

The provided algorithms are:

- Algorithm 1 and Algorithm 2: Bootstrapped DQN and Bootstrapped DQN + CERL. The changes needed for CERL are highlighted in yellow.
- Algorithm 3 and Algorithm 4: Ensemble SAC and Ensemble SAC + CERL. The changes needed for CERL are highlighted in yellow. For Ensemble SAC + CERL, we find it helpful to use huber loss with a threshold of 10 for the CERL auxiliary loss to prevent certain diverging ensemble members from affecting all other members. This is shown in the algorithm description. Also note that Clipped Double Q-learning (Fujimoto et al., 2018b) is used in practice but omitted in the algorithm description.

Note the above only describes the behavior of these algorithms during training. When evaluating the individual ensemble members' performance, these algorithms behave exactly the same as training time except that there are no learning updates. When evaluating the aggregate policies, we aggregate the policies of different ensemble members. For Bootstrapped DQN (+CERL), the policies are the greedy policies with respect to $\{Q_i\}_{i=1}^N$ (or the main heads $\{Q_i^i\}_{i=1}^N$ for CERL) and are aggregated via majority voting; specifically, for each visited state during evaluation, for each action in the action space, we count the number of ensemble members that select that action (i.e., votes) and we select the action with the most votes. Ties are broken randomly. For SAC we simply use the average of actions sampled from different policies. For example, if we visit state $s$ during evaluation, we will sample $a_i \sim \pi_i(\cdot|s)$ from each ensemble member, and then take the action $\frac{1}{N}\sum_{i=1}^N a_i$ in the environment.

# B    EXPERIMENTAL DETAILS

## B.1    ATARI

For Atari, we use the same set of 55 games as Agarwal et al. (2021). Following (Castro et al., 2018), the training process is divided into 200 iterations, each of which contains 1M frames. At the end of each iteration, the networks are frozen and evaluated with at least 500k frames. All Atari experiments use 5 seeds.

For each game, the human normalized scores (HNS) are computed as follows

$$\text{Score}_{\text{normalized}} = \frac{\text{Score}_{\text{Agent}} - \text{Score}_{\text{Random}}}{\text{Score}_{\text{Human}} - \text{Score}_{\text{Random}}}$$

where $\text{Score}_{\text{Agent}}$ is the raw score of the considered agent, and $\text{Score}_{\text{Human}}$ and $\text{Score}_{\text{Random}}$ are the raw scores of the human and the random agent respectively. The raw scores of an agent are calculated as the undiscounted evaluation returns averaged over the last 10 training iteration and 5 seeds. The scores of the human and random agents are taken from DQN Zoo (Quan & Ostrovski, 2020). To obtain results in Double DQN normalized scores, simply replace $\text{Score}_{\text{Human}}$ with $\text{Score}_{\text{DDQN}}$ in the above equations. $\text{Score}_{\text{DDQN}}$ is the raw score obtained by the Double DQN agent, averaged over the last 10 training iteration and 5 seeds.

When showing per-game improvements, we simply compute the difference in human-normalized scores. Per-game improvement bar plots (e.g., Figure 2 (top-right)) are shown in log-scale with a linear threshold at 0.1.

## B.2    MUJOCO

Each agent is trained for a total of 1M steps. Evaluation is performed every 20k steps with 30 evaluation episodes. When reporting the final performance in an environment, we average the last

---

**Algorithm 1** Bootstrapped DQN (training)

---

**Require:** total interaction steps $M$, ensemble size $N$, gradient update period $P$, target update period $T$, $N$ value functions $\{Q_i(s, a; \theta)\}_{i=1}^N$, $N$ target value functions $\{\bar{Q}_i(s, a; \bar{\theta})\}_{i=1}^N$, replay buffer $\mathcal{D}$

$t \leftarrow \text{True}$ $\quad\quad\quad\quad\quad\quad\quad\quad\quad\quad$ ▷ Terminal state indicator/whether to start a new episode

**for** $m \leftarrow 1$ **to** $M$ **do**
$\quad$ **if** $t = \text{True}$ **then** $\quad\quad\quad\quad\quad\quad\quad\quad\quad\quad\quad\quad\quad\quad\quad\quad\quad\quad$ ▷ New episode
$\quad\quad$ $s_m \leftarrow \text{reset(env)}$
$\quad\quad$ $k \sim \text{Uniform}(\{1, \dots, N\})$ $\quad\quad\quad\quad\quad\quad$ ▷ Randomly sample a member for acting
$\quad$ **else**
$\quad\quad$ $s_m \leftarrow s'_{m-1}$
$\quad$ **end if**
$\quad$ $a_m \leftarrow \arg\max_a Q_k(s_m, a; \theta)$
$\quad$ $a_m \sim \epsilon\text{-greedy}(a_m, m)$ $\quad\quad\quad\quad\quad$ ▷ Epsilon greedy policy with a decay schedule
$\quad$ $s'_m, r_m, t_m \leftarrow \text{step(env}, a_m)$
$\quad$ Add $(s_m, a_m, r_m, s'_m, t_m)$ to the shared replay buffer $\mathcal{D}$
$\quad$ **if** $m \bmod P = 0$ **then** $\quad\quad\quad\quad\quad\quad\quad\quad\quad\quad\quad\quad\quad\quad$ ▷ Gradient update
$\quad\quad$ Sample a transition $(s, a, r, s', t)$ from $\mathcal{D}$
$\quad\quad$ **for** $j \in \{1, \dots, N\}$ **do**
$\quad\quad\quad$ $a' \leftarrow \arg\max_{a'} Q_j(s', a'; \theta)$
$\quad\quad\quad$ $y_j \leftarrow \begin{cases} r + \gamma \bar{Q}_j(s', a'; \bar{\theta}) & \text{if } t = \text{False} \\ r & \text{if } t = \text{True} \end{cases}$ $\quad\quad$ ▷ Double DQN update
$\quad\quad$ **end for**
$\quad\quad$ $L(\theta) \leftarrow \sum_{i=1}^N \text{huber\_loss}(Q_i(s, a; \theta) - y_i)$
$\quad\quad$ $\delta\theta \leftarrow \nabla_\theta L(\theta)$
$\quad\quad$ $\delta\theta \leftarrow \text{scale\_grad}(\delta\theta)$ $\quad\quad\quad\quad$ ▷ Scale the gradients of the encoder(s). See Section C.2.
$\quad\quad$ $\theta \leftarrow \text{optimizer}(\theta, \delta\theta)$
$\quad$ **end if**
$\quad$ **if** $m \bmod T = 0$ **then**
$\quad\quad$ $\bar{\theta} \leftarrow \theta$ $\quad\quad\quad\quad\quad\quad\quad\quad\quad\quad\quad\quad\quad\quad\quad\quad\quad\quad\quad\quad$ ▷ Update target network
$\quad$ **end if**
**end for**

---

---

**Algorithm 2** Bootstrapped DQN + CERL (training)

---

**Require:** total interaction steps $M$, ensemble size $N$, gradient update period $P$, target update period $T$, $N \times N$ value functions $\{Q_i^j(s, a; \theta)\}_{i,j=1}^N$, $N \times N$ target value functions $\{\bar{Q}_i^j(s, a; \bar{\theta})\}_{i,j=1}^N$, replay buffer $\mathcal{D}$

$t \leftarrow$ True            ▷ Terminal state indicator/whether to start a new episode

**for** $m \leftarrow 1$ **to** $M$ **do**
 **if** $t =$ True **then**               ▷ New episode
  $s_m \leftarrow$ reset(env)
  $k \sim$ Uniform($\{1, \ldots, N\}$)       ▷ Randomly sample a member for acting
 **else**
  $s_m \leftarrow s'_{m-1}$
 **end if**
 $a_m \leftarrow \arg\max_a Q_k^k(s_m, a; \theta)$
 $a_m \sim \epsilon\text{-greedy}(a_m, m)$      ▷ Epsilon greedy policy with a decay schedule
 $s'_m, r_m, t_m \leftarrow$ step(env, $a_m$)
 Add $(s_m, a_m, r_m, s'_m, t_m)$ to the shared replay buffer $\mathcal{D}$
 **if** $m \mod P = 0$ **then**            ▷ Gradient update
  Sample a transition $(s, a, r, s', t)$ from $\mathcal{D}$
  **for** $j \in \{1, \ldots, N\}$ **do**
   $a' \leftarrow \arg\max_{a'} Q_j^j(s', a'; \theta)$

$$y_j \leftarrow \begin{cases} r + \gamma \bar{Q}_j^j(s', a'; \bar{\theta}) & \text{if } t = \text{False} \\ r & \text{if } t = \text{True} \end{cases} \qquad \triangleright \text{ Double DQN update}$$

  **end for**
  $L(\theta) \leftarrow \sum_{i=1}^N \sum_{j=1}^N \text{huber\_loss}(Q_i^j(s, a; \theta) - y_j)$
  $\delta\theta \leftarrow \nabla_\theta L(\theta)$
  $\delta\theta \leftarrow$ scale\_grad($\delta\theta$)    ▷ Scale the gradients of the encoder(s). See Section C.2.
  $\theta \leftarrow$ optimizer($\theta, \delta\theta$)
 **end if**
 **if** $m \mod T = 0$ **then**
  $\bar{\theta} \leftarrow \theta$                 ▷ Update target network
 **end if**
**end for**

---

---

**Algorithm 3** Ensemble SAC (training)

---

**Require:** total interaction steps $M$, ensemble size $N$, gradient update period $P$, target update
temperature $\tau$, $N$ value functions $\{Q_i(s,a;\theta)\}_{i=1}^N$, $N$ target value functions $\{\bar{Q}_i(s,a;\bar{\theta})\}_{i=1}^N$, $N$
policies $\{\pi_i(a|s;\phi)\}_{i=1}^N$, $N$ entropy temperatures $\{\alpha_i\}_{i=1}^N$, replay buffer $\mathcal{D}$
$t \leftarrow$ True                        $\triangleright$ Terminal state indicator/whether to start a new episode
**for** $m \leftarrow 1$ **to** $M$ **do**
   **if** $t =$ True **then**                        $\triangleright$ New episode
      $s_m \leftarrow$ reset(env)
      $k \sim$ Uniform($\{1, \ldots, N\}$)                        $\triangleright$ Randomly sample a member for acting
   **else**
      $s_m \leftarrow s'_{m-1}$
   **end if**
   $a_m \sim \pi_k(\cdot|s_m;\phi)$
   $s'_m, r_m, t_m \leftarrow$ step(env, $a_m$)
   Add $(s_m, a_m, r_m, s'_m, t_m)$ to the shared replay buffer $\mathcal{D}$
   **if** $m \bmod P = 0$ **then**                        $\triangleright$ Gradient update
      Sample a transition $(s, a, r, s', t)$ from $\mathcal{D}$
      **for** $j \in \{1, \ldots, N\}$ **do**
         $a' \sim \pi_j(\cdot|s';\phi)$
         $y_j \leftarrow \begin{cases} r + \gamma\bar{Q}_j(s', a';\bar{\theta}) & \text{if } t = \text{False} \\ r & \text{if } t = \text{True} \end{cases}$
         $a_j(\phi) \sim \pi_j(\cdot|s;\phi)$                        $\triangleright$ Note with the parametrization trick, $a_j$ depends on $\phi$
      **end for**
      $L_{\text{critic}}(\theta) \leftarrow \sum_{i=1}^N (Q_i(s,a;\theta) - y_i)^2$
      $\delta\theta \leftarrow \nabla_\theta L_{\text{critic}}(\theta)$ and $\theta \leftarrow$ optimizer($\theta, \delta\theta$)                        $\triangleright$ Update critic
      $L_{\text{actor}}(\phi) \leftarrow -\sum_{i=1}^N (Q_i(s, a_i(\phi);\theta) - \alpha_i \log \pi_i(a_i(\phi)|s;\phi))$
      $\delta\phi \leftarrow \nabla_\phi L_{\text{actor}}(\phi)$ and $\phi \leftarrow$ optimizer($\theta, \delta\phi$)                        $\triangleright$ Update actor
      Update the entropy temperatures $\{\alpha_i\}$ for each member as in standard SAC
   **end if**
   $\bar{\theta} \leftarrow (1 - \tau)\bar{\theta} + \tau\theta$                        $\triangleright$ Update target network
**end for**

---

---

**Algorithm 4** Ensemble SAC + CERL (training)

---

**Require:** interaction steps $M$, ensemble size $N$, gradient update period $P$, target update temperature $\tau$, $N \times N$ value functions $\{Q_i^j(s, a; \theta)\}_{i,j=1}^N$, $N \times N$ target value functions $\{\bar{Q}_i^j(s, a; \bar{\theta})\}_{i,j=1}^N$, $N$ entropy temperatures $\{\alpha_i\}_{i=1}^N$, replay buffer $\mathcal{D}$

$t \leftarrow$ True             ▷ Terminal state indicator/whether to start a new episode

**for** $m \leftarrow 1$ **to** $M$ **do**
    **if** $t =$ True **then**             ▷ New episode
        $s_m \leftarrow \text{reset(env)}$
        $k \sim \text{Uniform}(\{1, \ldots, N\})$             ▷ Randomly sample a member for acting
    **else**
        $s_m \leftarrow s'_{m-1}$
    **end if**
    $a_m \sim \pi_k(\cdot | s_m; \phi)$
    $s'_m, r_m, t_m \leftarrow \text{step(env}, a_m)$
    Add $(s_m, a_m, r_m, s'_m, t_m)$ to the shared replay buffer $\mathcal{D}$
    **if** $m \bmod P = 0$ **then**             ▷ Gradient update
        Sample a transition $(s, a, r, s', t)$ from $\mathcal{D}$
        **for** $j \in \{1, \ldots, N\}$ **do**
            $a' \sim \pi_j(\cdot | s'; \phi)$

$$y_j \leftarrow \begin{cases} r + \gamma \bar{Q}_j^j(s', a'; \bar{\theta}) & \text{if } t = \text{False} \\ r & \text{if } t = \text{True} \end{cases}$$

            $a_j(\phi) \sim \pi_j(\cdot | s; \phi)$        ▷ Note with the parametrization trick, $a_j$ depends on $\phi$
        **end for**

$$L_{\text{critic}}(\theta) \leftarrow \sum_{i=1}^N \sum_{j=1, j \neq i}^N \text{huber\_loss}(Q_i^j(s, a; \theta) - y_j) + \sum_{i=1}^N (Q_i^i(s, a; \theta) - y_i)^2$$

        $\delta\theta \leftarrow \nabla_\theta L_{\text{critic}}(\theta)$ and $\theta \leftarrow \text{optimizer}(\theta, \delta\theta)$        ▷ Update critic

$$L_{\text{actor}}(\phi) \leftarrow -\sum_{i=1}^N (Q_i^i(s, a_i(\phi); \theta) - \alpha_i \log \pi_i(a_i(\phi) | s; \phi))$$

        $\delta\phi \leftarrow \nabla_\phi L_{\text{actor}}(\phi)$ and $\phi \leftarrow \text{optimizer}(\theta, \delta\phi)$        ▷ Update actor
        Update the entropy temperatures $\{\alpha_i\}$ for each member as in standard SAC
    **end if**
    $\bar{\theta} \leftarrow (1 - \tau)\bar{\theta} + \tau\theta$             ▷ Update target network
**end for**

---

5 evaluation results at the end of training. All MuJoCo experiments use 30 seeds except for the 10%-tandem experiments, where we use 10 seeds.

### B.3  $p\%$-TANDEM EXPERIMENTS

Taking Bootstrapped DQN as an example, the easiest way to understand the experimental setup is by noticing that setting $p = 50\%$ recovers the standard Bootstrapped DQN with 2 ensemble members. The only differences of $p\%$-tandem to Bootstrapped DQN with $N = 2$ are (1) for each training episode, instead of sampling each agent for acting with the same $50\%$ probability, we sample the active agent with probability $1 - p\%$ and the passive agent with probability $p\%$ and (2) we record the performances of both the active and passive agents separately during evaluation.

## C  IMPLEMENTATION DETAILS

### C.1  IMPLEMENTATION AND HYPERPARAMETERS

The SAC and ensemble SAC algorithms are built on JAXRL (Kostrikov, 2021) and use the default hyperparameters. In the actor implementation, we use tanh instead of hard clipping for the `log_std` parameter for better stability. Double DQN and Bootstrapped DQN are implemented using the Dopamine (Castro et al., 2018) framework in JAX (Bradbury et al., 2018). As our work heavily refers to Osband et al. (2016) and Ostrovski et al. (2021), we mostly follow the hyperparameter setup in DQN Zoo (Quan & Ostrovski, 2020) because those are more similar to the ones used in Osband et al. (2016) and Ostrovski et al. (2021). We do not directly build on DQN Zoo mainly because the implementation in Dopamine is more efficient. The hyperparameters for Double DQN are listed in Table 1. Note one "agent steps" corresponds to 4 environment frames due to the use of action repetitions/frame-skip.

Bootstrapped DQN reuses all the hyperparameters of Double DQN, with two additional hyperparameters: ensemble size $N$ and the number of shared bottom layers $L$. We set $N = 10$ and $L = 0$ by default.

As mentioned in Appendix A, for Ensemble SAC + CERL, we find it helpful to use huber loss with a threshold of 10 for the CERL auxiliary loss to prevent certain diverging ensemble members from affecting all other members. The main head still uses the regular MSE loss so the use of Huber loss only affects the auxiliary task.

### C.2  GRADIENT SCALING

Following Osband et al. (2016), we scale the gradients of the encoder(s) to ensure that the magnitude of the gradients does not increase with the number of heads. For example, in CERL, the gradients of the encoders $f_i$ will be divided by $N$, which is the number of heads on top of this encoder.

### C.3  COMPUTATIONAL COSTS

On a server with NVIDIA RTX8000 GPU and AMD EPYC 7502 CPU, each seed of our implementation of Double DQN, Bootstrapped DQN ($N = 10$, share 0 layers), and Bootstrapped DQN ($N = 10$, share 3 layers) take roughly 2, 3.5, and 3 days respectively for 200M frames. Applying CERL does not noticeably increase the wall clock time in our experiments. Running 10 seeds of SAC, Ensemble SAC, and Ensemble SAC + CERL with $N = 10$ on Humanoid-v4 takes roughly 5, 12.5, 13 hours respectively.

## D  ADDITIONAL RESULTS

### D.1  10%-TANDEM EXPERIMENTS FOR CONTINUOUS CONTROL TASKS

In Figure 8 we show the results of the 10%-tandem experiment in MuJoCo tasks, which shares the same patterns as the one in Atari. In MuJoCo tasks, the performance gap between the active and the passive agent is larger than that between Ensemble SAC (indiv.) and SAC. This might be because

Table 1: Hyperparameters for Double DQN

| Parameter | Value |
|---|---|
| Gray-scaling | True |
| Observation down-sampling | $84 \times 84$ |
| Frames stacked | 4 |
| Action repetitions | 4 |
| Sticky actions | False |
| Reward clipping | $[-1, 1]$ |
| Terminal on loss of life | False |
| Max frames per episode | 108k |
| $Q$ update rule | Double DQN |
| Discount factor | 0.99 |
| Minibatch size | 32 |
| Replay buffer size | $10^6$ |
| Optimizer | RMSProp |
| Optimizer: learning rate | 0.00025 |
| Optimizer: RMSProp decay | 0.95 |
| Optimizer: RMSProp centered | True |
| Optimizer: $\epsilon$ | $1/32^2$ |
| Huber loss | True |
| Evaluation frames | 500k |
| Evaluation period in frames | 1M |
| Min replay size for sampling | 50000 |
| Gradient update period in agent steps | 4 |
| Target network update period in agent steps | 30000 |
| Exploration: $\epsilon$ during training | $1.0 \rightarrow 0.01$ |
| Exploration: $\epsilon$ decay period in agent steps | 16M |
| Exploration: $\epsilon$ during evaluation | 0.01 |
| Q network: channels | 32, 64, 64 |
| Q network: filter size | $8 \times 8, 4 \times 4, 3 \times 3$ |
| Q network: stride | 4, 2, 1 |
| Q network: hidden units | 512 |
| Q network: padding type | valid convolution |
| Q network: share bias across action heads | True |
| Q network: initialization | See Quan & Ostrovski (2020) |

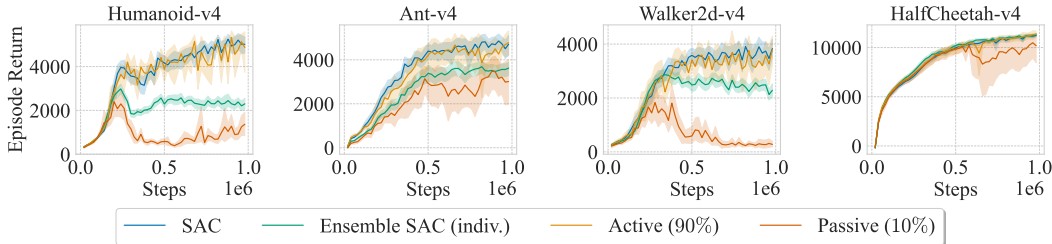

Figure 8: 10%-tandem experiments in MuJoCo tasks with a replay buffer size of 200k. Results are aggregated over 10 seeds for the active and passive agents and 30 seeds for others. Shaded areas show 95% CIs.

Ensemble SAC (indiv.) provides better exploration and partially compensates for the performance loss due to challenging off-policy learning.

## D.2   Per-game results

### D.2.1   Per-game learning curves

Figure 9 shows per-game comparisons between Double DQN, vanilla Bootstrapped DQN, and CERL with policy aggregation. Figure 10 shows similar comparisons without policy aggregation. Figure 11 shows the per-game results for the 10%-tandem experiment.

### D.2.2   Per-game analysis of the benefits of majority voting

In Figure 12 we show the performance gap between Bootstrapped DQN (agg.) and Bootstrapped DQN (indiv.) in each game. As shown in the results, majority voting provides significant performance gains in almost all games.

### D.2.3   Per-game performance of CERL

Per-game performance of CERL is shown in Figure 10 and Figure 9. We also show per-game improvements of CERL over Bootstrapped DQN in Figure 13.

## D.3   Additional analysis of the curse of diversity

### D.3.1   Over-sampling self-generated data in the training batches

Even though we cannot control the proportion of self-generated data in the replay buffer for each ensemble member, it is possible to control this proportion in the *training batches*. In this experiment, for each ensemble member, with 50% probability we only sample the training batches from self-generated transitions; otherwise, we sample uniformly from all the data as usual. This ensures the proportion of self-generated data in the training batches for each ensemble member is at least 50%.

Figure 14 shows the effects of over-sampling self-generated data in the training batches for each ensemble member. As can be seen, this technique does not mitigate the performance loss relative to single-agent Double DQN.

### D.3.2   Episode termination condition

In Atari, we have the option to terminate an episode when a life is lost, or only when the game is over. The former option may help the agent quickly learn the significance of death. In the context of ensemble-based exploration, shorter episodes mean it is less likely that a single ensemble will dominate the environment interaction for a long stretch of time, during which other ensemble members perform no interaction at all.

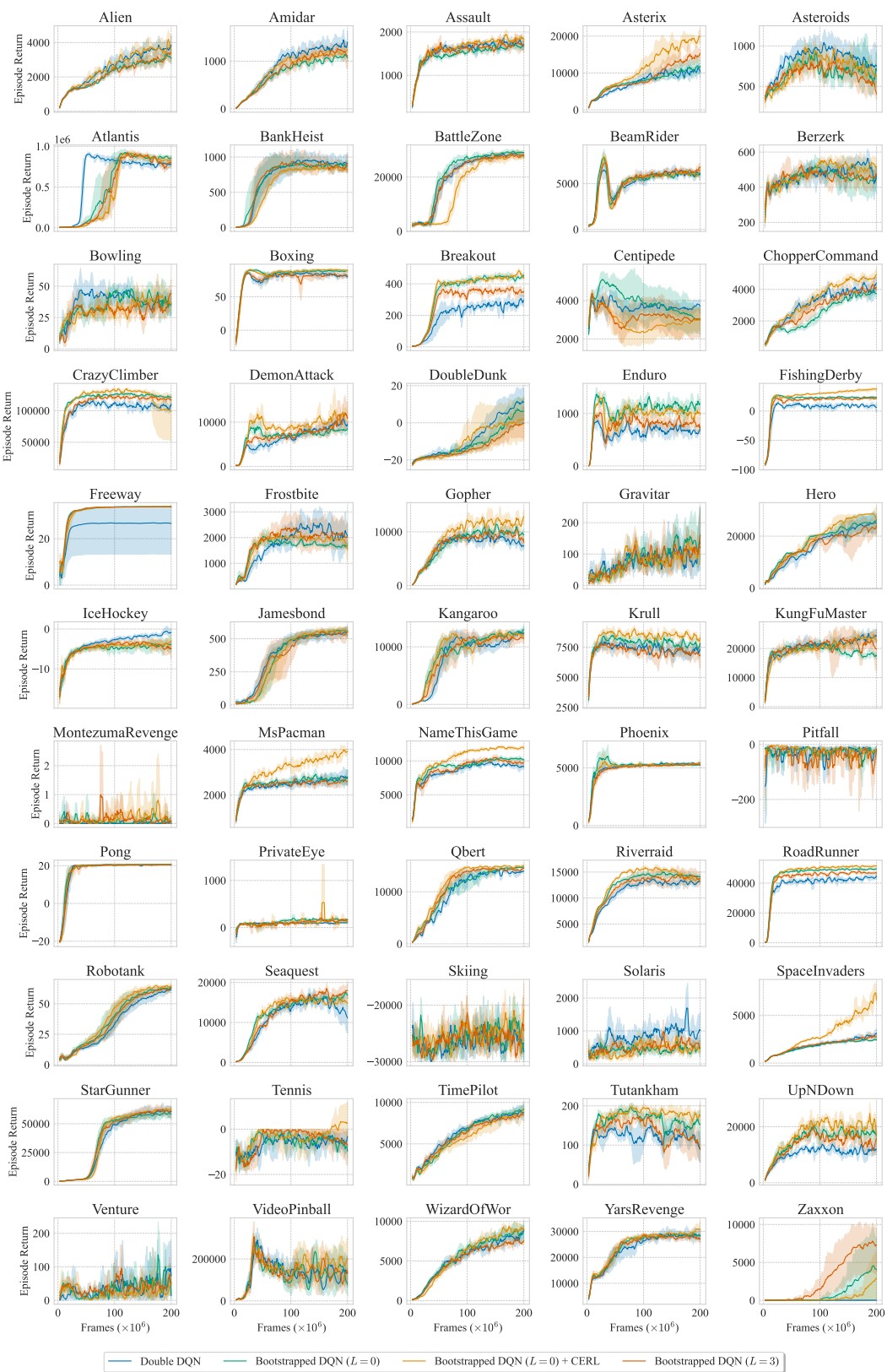

Figure 9: Per-game comparisons between Double, vanilla Bootstrapped DQN, and CERL with voting. All results are aggregated over 5 seeds. Shaded areas show 95% bootstrapped CIs. The learning curves are smoothed with a sliding window of 5 iterations.

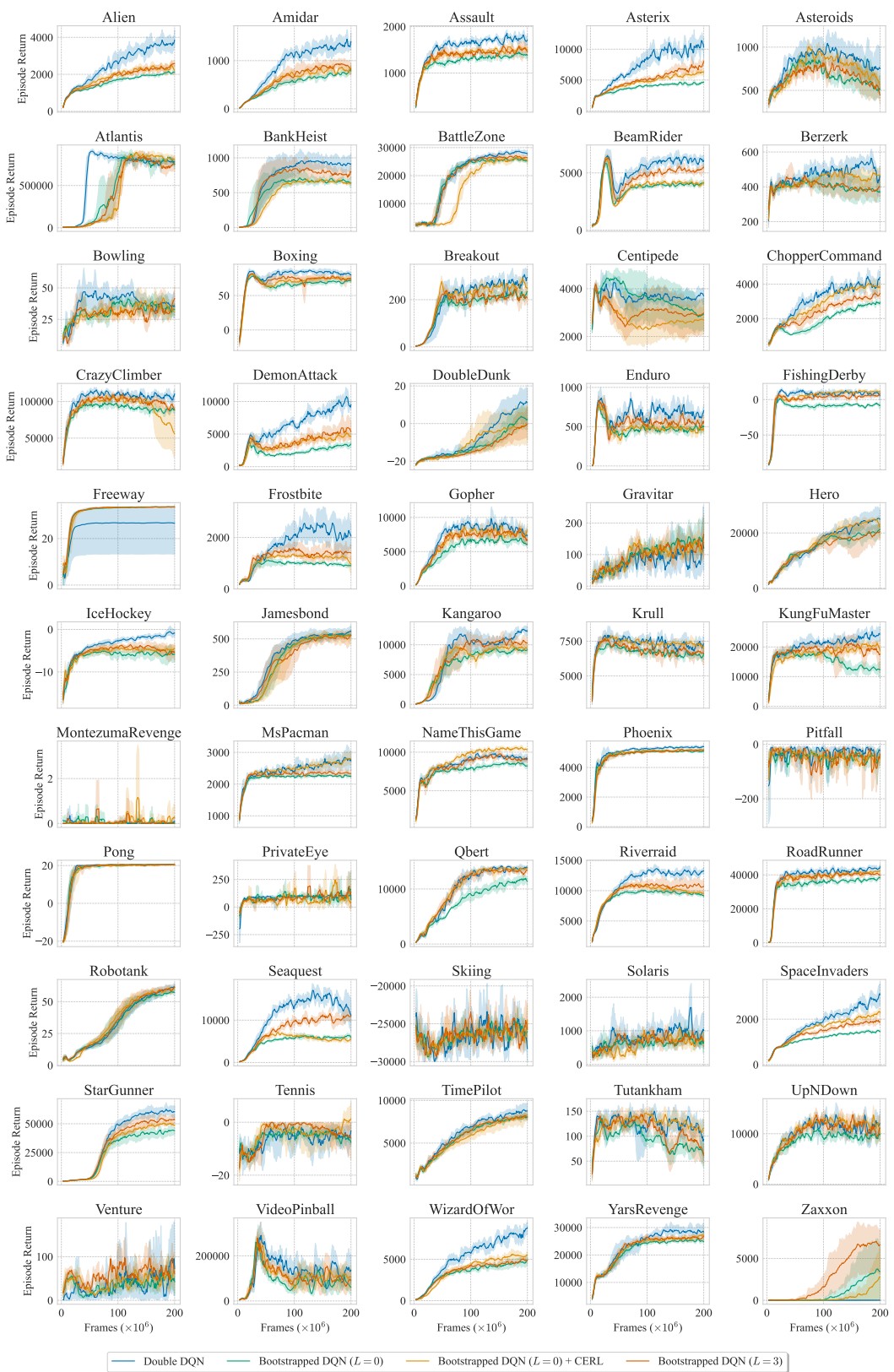

Figure 10: Per-game comparisons between Double, vanilla Bootstrapped DQN, and CERL without voting. All results are aggregated over 5 seeds. Shaded areas show 95% bootstrapped CIs. The learning curves are smoothed with a sliding window of 5 iterations.

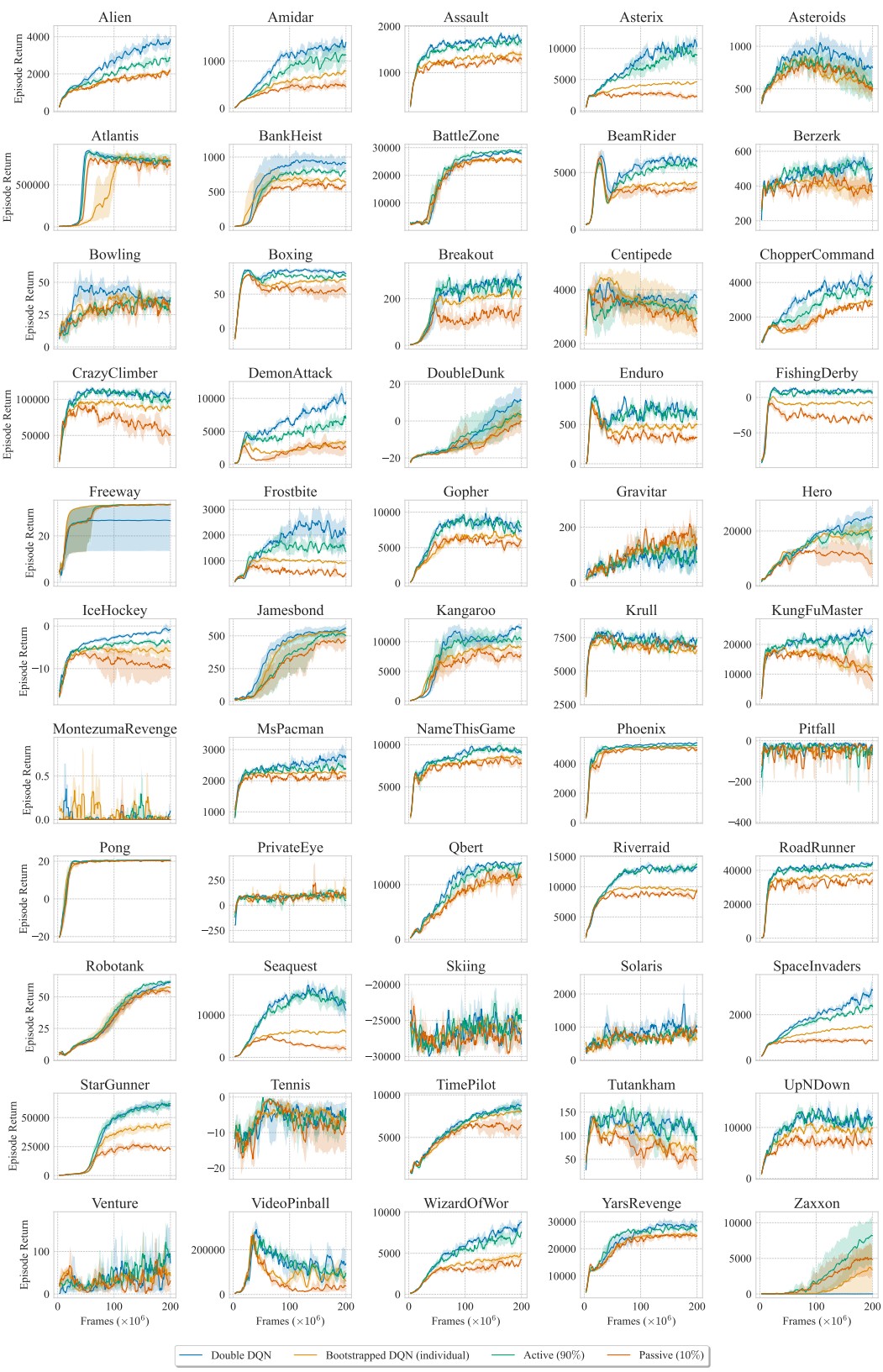

Figure 11: Per-game comparisons of the 10%-tandem experiment. All results are aggregated over 5 seeds. Shaded areas show 95% bootstrapped CIs. The learning curves are smoothed with a sliding window of 5 iterations.

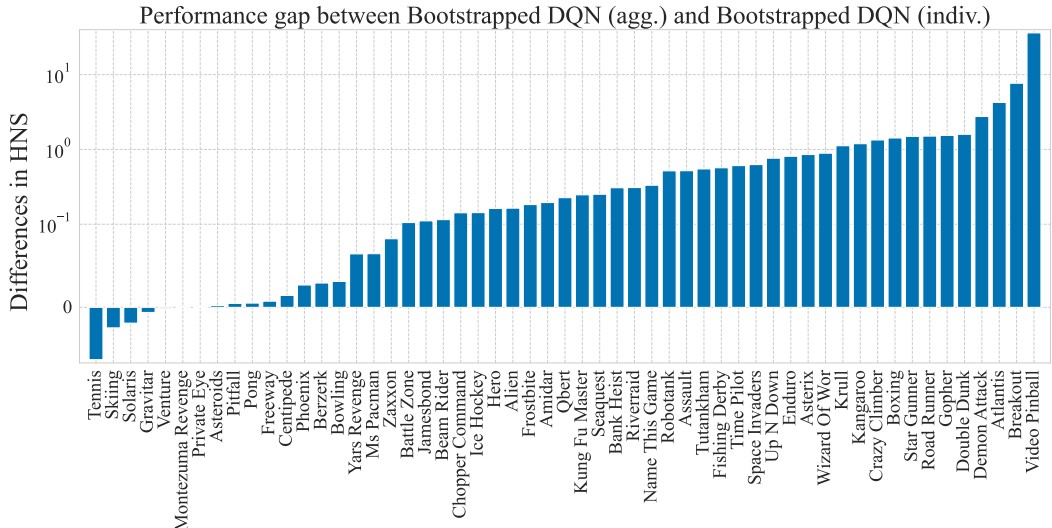

Figure 12: Per-game performance gap between Bootstrapped DQN (agg.) and Bootstrapped DQN (indiv.) in HNS.

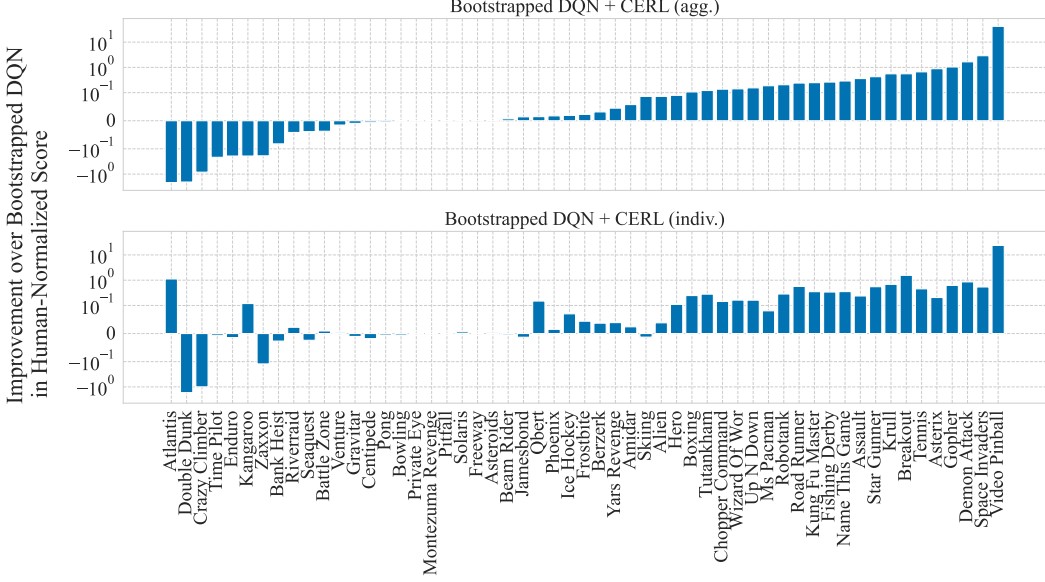

Figure 13: Per-game improvements of Bootstrapped DQN + CERL over Bootstrapped DQN in HNS, with and without policy aggregation.

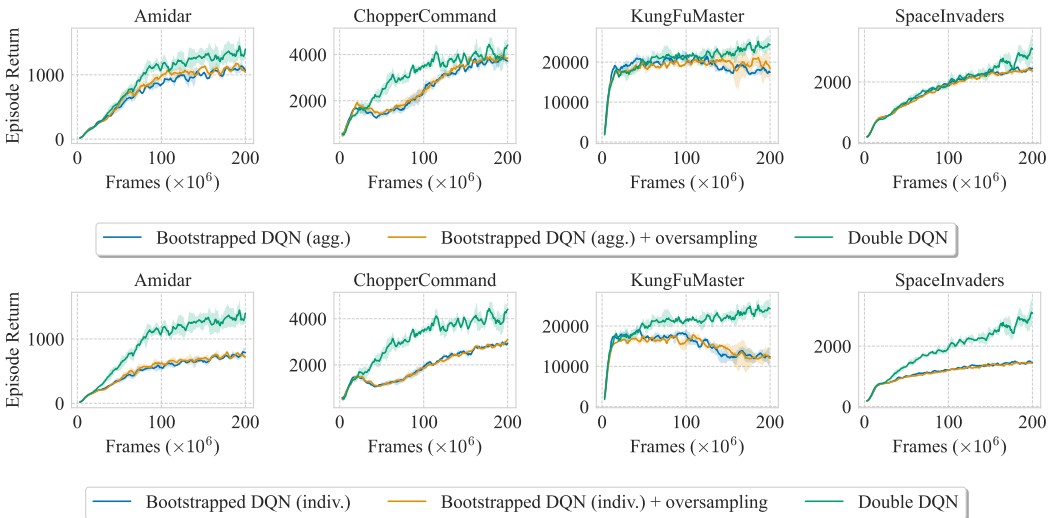

Figure 14: Effects of oversampling self-generated data in the training batches. (**top**) Performance with voting. (**bottom**) Performance without voting. All results are aggregated over 5 seeds. Shaded areas show 95% bootstrapped CIs. The learning curves are smoothed with a sliding window of 5 iterations.

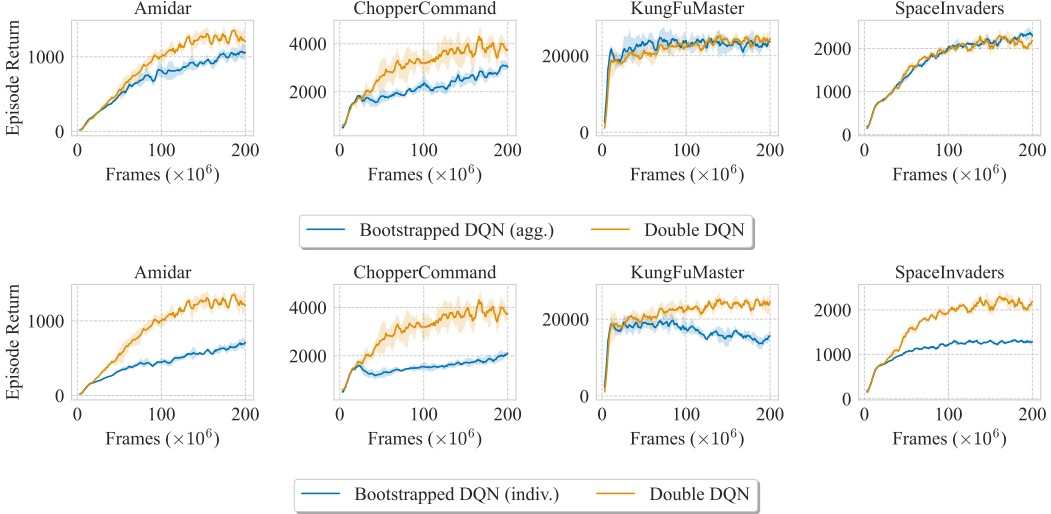

Figure 15: Results with the episode termination condition set to life loss. (**top**) Performance with voting. (**bottom**) Performance without voting. All results are aggregated over 5 seeds. Shaded areas show 95% bootstrapped CIs. The learning curves are smoothed with a sliding window of 5 iterations.

Our work follows the recommendation of Machado et al. (2017) and only terminates an episode when the game is over. In Figure 15 we show the performance of Bootstrapped DQN and Double DQN when we set the termination condition to life loss. As can be seen, the curse of diversity still remains.

### D.3.3 DISTRIBUTIONAL RL

Recent work (Agarwal et al., 2020) has shown that the QR-DQN algorithm (Dabney et al., 2017) can be more effective than DQN in offline RL. In Figure 16 we test a variant of Bootstrapped DQN where we train an ensemble of QR-DQN agents instead of Double DQN agents (named Bootstrapped QR-DQN). We use the implementation and default hyperparameters of QR-DQN in Dopamine (Castro

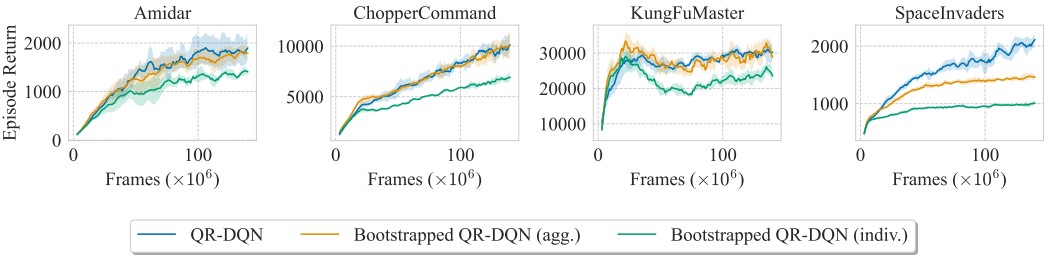

Figure 16: Performance of Bootstrapped DQN when using QR-DQN as the base algorithm. All results are aggregated over 5 seeds. Shaded areas show 95% bootstrapped CIs. The learning curves are smoothed with a sliding window of 5 iterations.

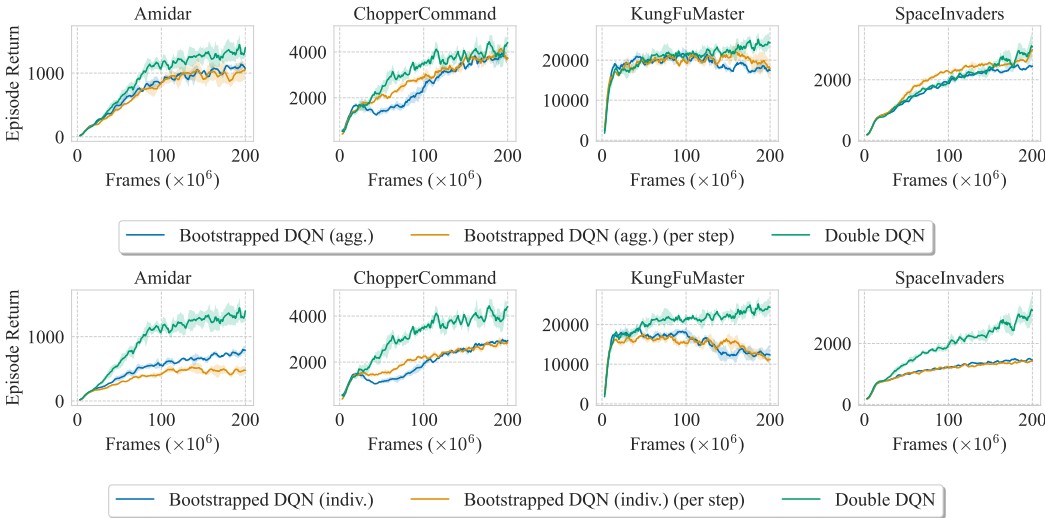

Figure 17: Effects of switching ensemble members on a per-step basis instead of per-episode. (**top**) Performance with voting. (**bottom**) Performance without voting. All results are aggregated over 5 seeds. Shaded areas show 95% bootstrapped CIs. The learning curves are smoothed with a sliding window of 5 iterations.

et al., 2018) except that we do not use prioritized experience replay (Schaul et al., 2015) as it is unclear whether it is appropriate to prioritize transitions based on the TD error of the *entire ensemble*.

As shown in Figure 16, there exists a significant performance gap between QR-DQN and Bootstrapped QR-DQN. This suggests that the curse of diversity is also present in distributional RL.

### D.3.4 SWITCHING ENSEMBLE MEMBERS ON A PER-STEP BASIS

The original Bootstrapped DQN switches the ensemble member used for acting at the start of each episode. In Figure 17, we test a variant of Bootstrapped DQN where we switch the ensemble member for acting *on a per-step basis*. Note that this no longer allows "temporally extended exploration" (Osband et al., 2016) which is the original motivation of Bootstrapped DQN, but it might mitigate the off-policy learning issue as it allows all ensemble members (as opposed to just one of them) to generate data within the period of an episode.

As shown in Figure 17, switching ensemble members more frequently does not mitigate the curse of diversity. This is not surprising as it does not change the proportion of self-generated data for each ensemble member in the replay buffer.

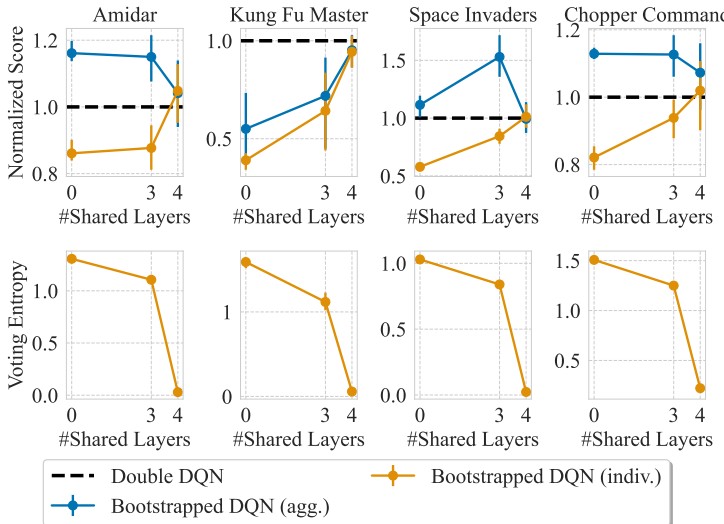

Figure 18: The effects of varying the number of shared layers when using a replay buffer of 4M transitions. The top row shows Double DQN normalized scores. The bottom row shows the entropy of the normalized vote distributions. Error bars show 95% bootstrapped CIs over 5 seeds.

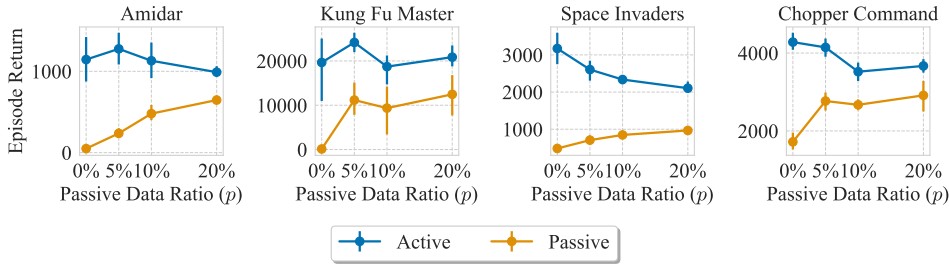

Figure 19: The effects of varying $p$ in the $p\%$-tandem experiment. Error bars show 95% bootstrapped CIs over 5 seeds.

### D.3.5 LAYER SHARING EXPERIMENT WITH A LARGER REPLAY BUFFER

In Figure 18 we repeat the layer-sharing experiment in Section 3.3 but with a replay buffer of 4M transitions. This allows Bootstrapped DQN (agg.) to outperform Double DQN in three games. However, the trade-off between the advantages and disadvantages of diversity remains: as we increase the number of shared layers and hence reduce diversity,

- The curse of diversity, i.e., the gap between Bootstrapped DQN (indiv.) and Double DQN reduces;

- The performance gain we get from majority voting, i.e. the gap between Bootstrapped DQN (agg.) and Bootstrapped DQN (indiv.), also reduces. An exception is Space Invaders, where the gap seems to slightly increase when $L$ increases from 0 to 3, which might be related to certain properties of this game.

### D.3.6 DIFFERENT LEVELS OF PASSIVITY FOR THE $p\%$-TANDEM EXPERIMENT

In Figure 19 we show the effect of varying $p$ in the $p\%$-tandem experiment. As expected, increasing $p$ (i.e., reducing the degree of passivity) reduces the performance gap between the active and passive agents.

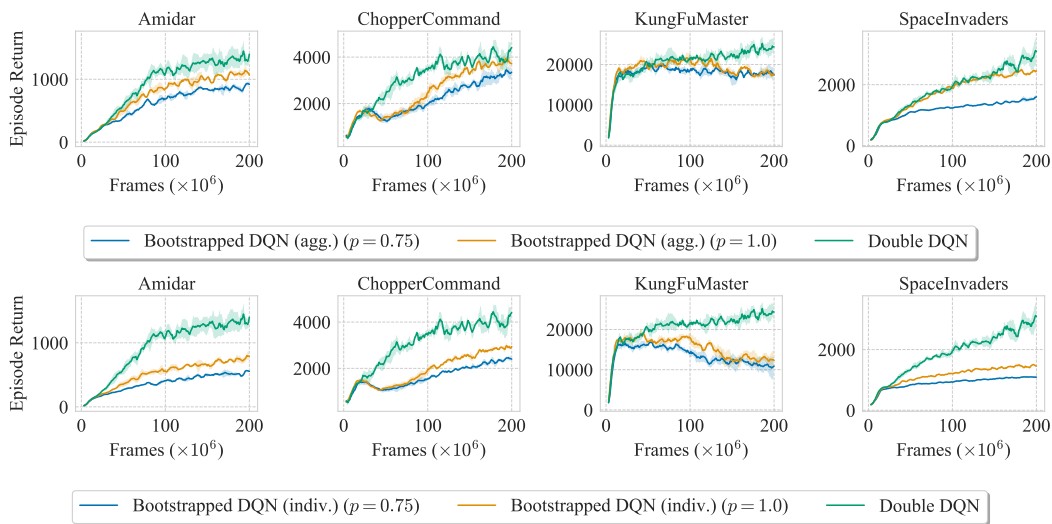

Figure 20: Effects of data bootstrapping. $p$ refers to the probability of masking out a certain sample for each ensemble member. Performance with voting. (**bottom**) Performance without voting. All results are aggregated over 5 seeds. Shaded areas show $95\%$ bootstrapped CIs. The learning curves are smoothed with a sliding window of 5 iterations.

### D.3.7 DATA BOOTSTRAPPING

As mentioned in Section 2 and Appendix A, we do not use data bootstrapping in Bootstrapped DQN (i.e., we set the masking probability $p$ to 1.0), as Osband et al. (2016) does not find it to be useful in Atari games. In Figure 20 we probe the effect of data bootstrapping on performance, with bootstrap masking probability $p = 0.75$ (see Osband et al. (2016) for how $p$ is used to approximate bootstrapping). As shown in the results, data bootstrapping damages performance on the environments we tested. This is not surprising because masking out samples essentially reduces the number of transitions each member has access to. It also effectively reduces the batch size. Besides, the fact that bootstrapping promotes diversity can exacerbate the curse of diversity. However, it is difficult to attribute the precise cause of the damaged performance.

### D.4 ADDITIONAL ANALYSIS OF CERL

#### D.4.1 ENSEMBLE SIZE ABLATION

In Figure 21 we test different ensemble sizes for CERL. With CERL, we see a clear trend that increasing the ensemble size gives better performance, though it saturates at $N = 10$ for two environments.

#### D.4.2 AN ALTERNATIVE DESIGN OF CERL

We consider the following alternative update rule for CERL:

$$Q_i^j(s,a) \leftarrow r + \gamma \bar{Q}_i^j(s', a_j'), \quad \text{for } i = 1, \dots, N, \quad \text{for } j = 1, \dots, N \quad (2)$$

where $\bar{Q}_i^j$ is the target network for $Q_i^j$ and $a_j' \sim \pi_j(\cdot|s')$. The only difference between the original CERL is that the original CERL uses $\bar{Q}_j^j$ on the right-hand side while here we use $\bar{Q}_i^j$ on the right-hand side. Since the next action still comes from $\pi_j$, this update rule is still trying to make the $j$-th head of ensemble $i$ learn the value functions of ensemble member $j$, but in a different way. Specifically, each auxiliary head uses itself to compute the TD target instead of other ensemble members' main heads. We refer to this variant as CERL (self-target).

In Figure 22 we show the performance of this variant on 4 games. As can be seen, it performs very well. However, in our preliminary experiments in MuJoCo tasks with Ensemble SAC, this variant

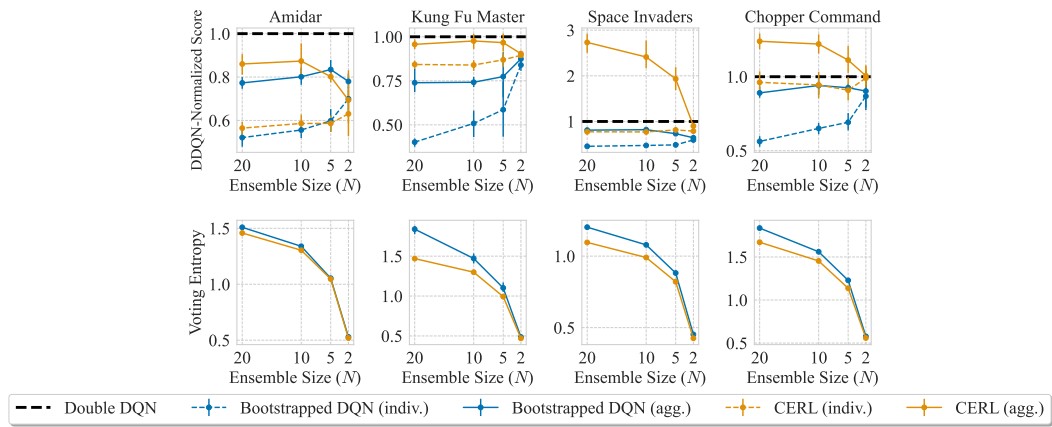

Figure 21: Impact of ensemble size on Bootstrapped DQN and CERL, with or without majority voting. Shaded areas show $95\%$ bootstrapped CIs over 5 seeds.

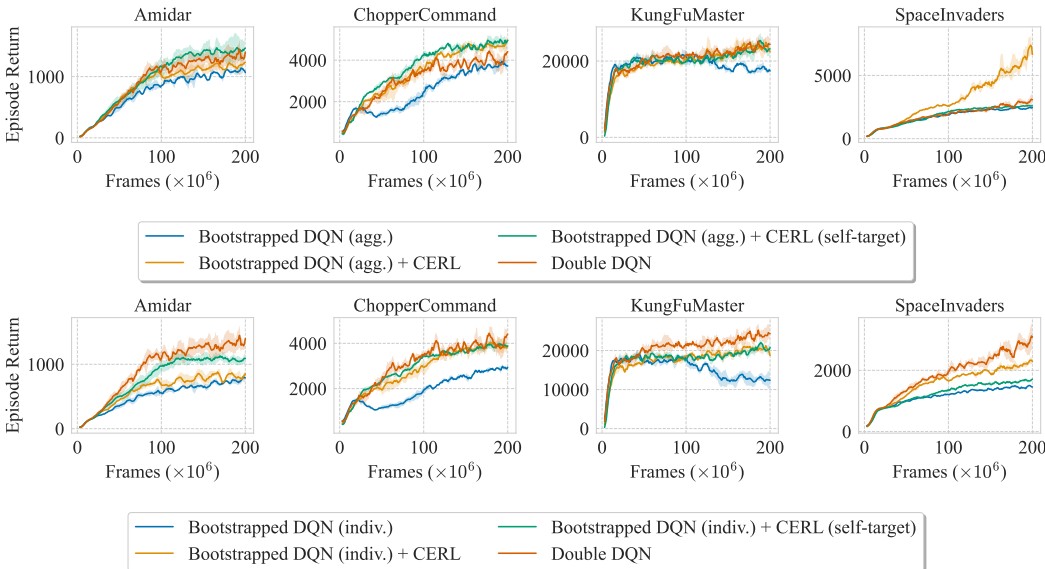

Figure 22: Performance of CERL (self-target) in 4 Atari games. (**top**) Performance with voting. (**bottom**) Performance without voting. Shaded areas show $95\%$ bootstrapped CIs over 5 seeds. The learning curves are smoothed with a sliding window of 5 iterations.

leads to instability in learning and we do not have a good explanation at this moment. Also, this variant requires additional forward passes for SAC (note we need to evaluate $\bar{Q}_i^j(s', a_j')$ for each $i$ and $j$).

### D.4.3 COMBINING CERL WITH ENCODER SHARING

Even though our original motivation is to obtain the representation learning effect of encoder sharing without actually sharing the encoders, we comment that it is possible to *use CERL with encoder sharing*. This results in a hierarchical architecture where the network "branches" twice near the output. In Figure 23 we apply CERL to Bootstrapped DQN ($L = 3$). Unfortunately, this provides almost no improvement over Bootstrapped DQN ($L = 3$). We conjecture that there are two reasons for this result. First, both methods shape the representations via jointly learning multiple value functions and their effects will likely overlap. Second, sharing layers reduces the diversity, and thus the learning signals from CERL is less informative.

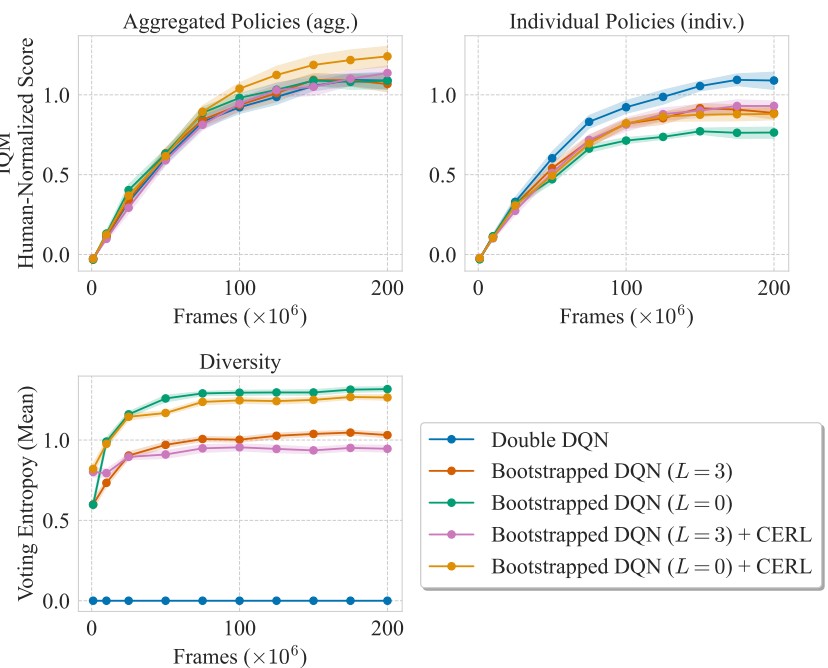

Figure 23: Effect of combining CERL and encoder sharing. Shaded areas show $95\%$ bootstrapped CIs over 5 seeds.

#### D.4.4 NUMBER OF PARAMETERS

As mentioned in the main text, applying CERL to Bootstrapped DQN ($N = 10$) without network sharing increases the number of parameters by no more than $5\%$. Further, these parameters are used for auxiliary tasks and thus do not affect the capacity of the part of the network that actually predicts the value functions. Though it is extremely unlikely that CERL's improvements come from increased parameters, for completeness we also test Bootstrapped DQN without CERL but with more parameters. We do so by increasing the output size of the penultimate layer such that the number of increased parameters is roughly the same as that introduced by CERL. As shown in Figure 24, this has almost no impact on performance.

#### D.4.5 OTHER REPRESENTATION LEARNING METHODS: A PRELIMINARY INVESTIGATION

The success of CERL suggests that representation learning in general may be promising for mitigating the curse of diversity. We perform a preliminary investigation with MICo (Castro et al., 2021) and multi-horizon auxiliary task (MH) (Fedus et al., 2019) in 55 Atari games. MICo is a metric-based method that explicitly shapes the representations, while MH does so implicitly by learning value functions of different horizons as an auxiliary task.

For the multi-horizon auxiliary task implementation, we jointly learn $K = 10$ value functions with $\{\gamma\}_{i=1}^{K}$, where $\gamma_i = 1 - \frac{1}{i \cdot (H_{\max}/K)}$, where $H_{\max} = 100$ leading to $\gamma_K = 0.99$. The architecture change involved is the same as that for CERL. Only the heads that correspond to the longest horizon $\gamma_K$ are used for acting. For MICo, we follow the author implementation[1] with $\beta = 0.1$. We search the MICo weight coefficient $\alpha$ in $\{0.01, 0.1, 0.5\}$ on the four Atari games we used in the main text based on the performance of Bootstrapped DQN + MICo. This results in the final selection of $\alpha = 0.01$.

We show the aggregate performance of Bootstrapped DQN + MH and Bootstrapped DQN + MICo in Figure 25 and Figure 26 respectively. Per-game results are summarized in Figure 27 and Figure 28 respectively. As these methods are also applicable to single-agent methods, we also show the performance of Double DQN + MH and Double DQN + MICo for completeness. As shown in

---

[1] https://github.com/google-research/google-research/tree/master/mico

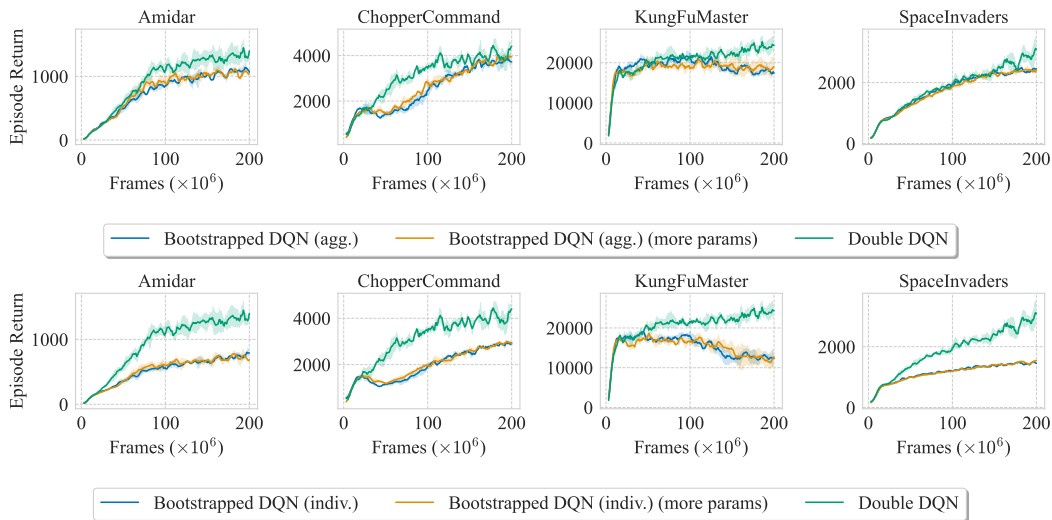

Figure 24: Bootstrapped DQN with more parameters. (**top**) Performance with voting. (**bottom**) Performance without voting. Shaded areas show $95\%$ bootstrapped CIs over $5$ seeds. The learning curves are smoothed with a sliding window of $5$ iterations.

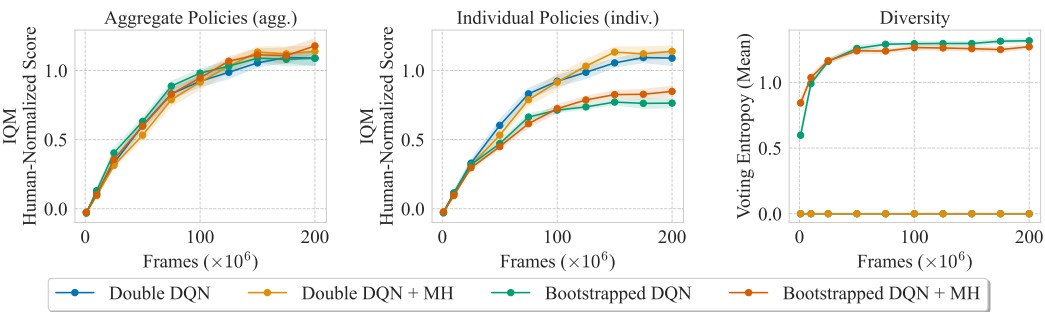

Figure 25: Comparison between Double DQN, Double DQN + MH, Bootstrapped DQN, and Bootstrapped DQN + MH in Atari. Results are aggregated over $55$ games and $5$ seeds. We show the performance of the agg. and indiv. versions of each ensemble algorithm in the top left and top middle plots respectively. Shaded areas show $95\%$ bootstrapped CIs.

these results, these methods do not provide a clear improvement to Bootstrapped DQN (indiv.) and Bootstrapped DQN (agg.) as CERL does. Note that in the result we find that MICo damages the performance of Double DQN. This is likely because our hyperparameter setup (largely based on DQN Zoo (Quan & Ostrovski, 2020), as mentioned in Appendix C) is very from those used in the original MICo paper, which is based on the setup used in Dopamine (Castro et al., 2018).

We emphasize that these are preliminary investigations, and the results may vary a lot based on the base algorithm, hyperparameters, and environments. A thorough analysis of what types of representations are most suited for addressing the curse of diversity is left for future work.

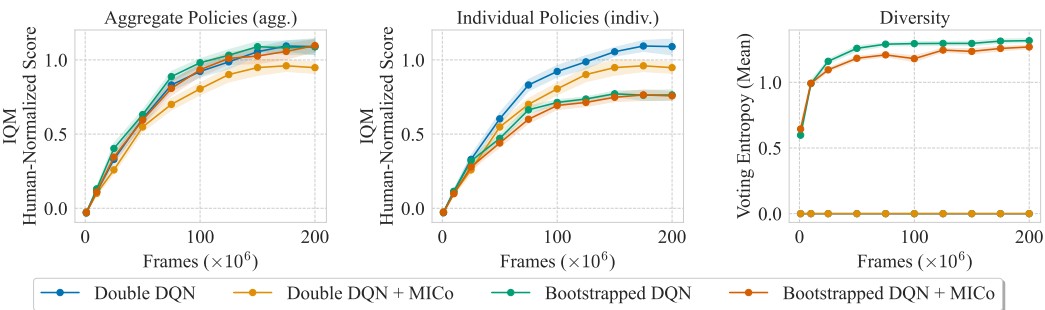

Figure 26: Comparison between Double DQN, Double DQN + MICo, Bootstrapped DQN, and Bootstrapped DQN + MICo in Atari. Results are aggregated over 55 games and 5 seeds. We show the performance of the agg. and indiv. versions of each ensemble algorithm in the top left and top middle plots respectively. Shaded areas show 95% bootstrapped CIs.

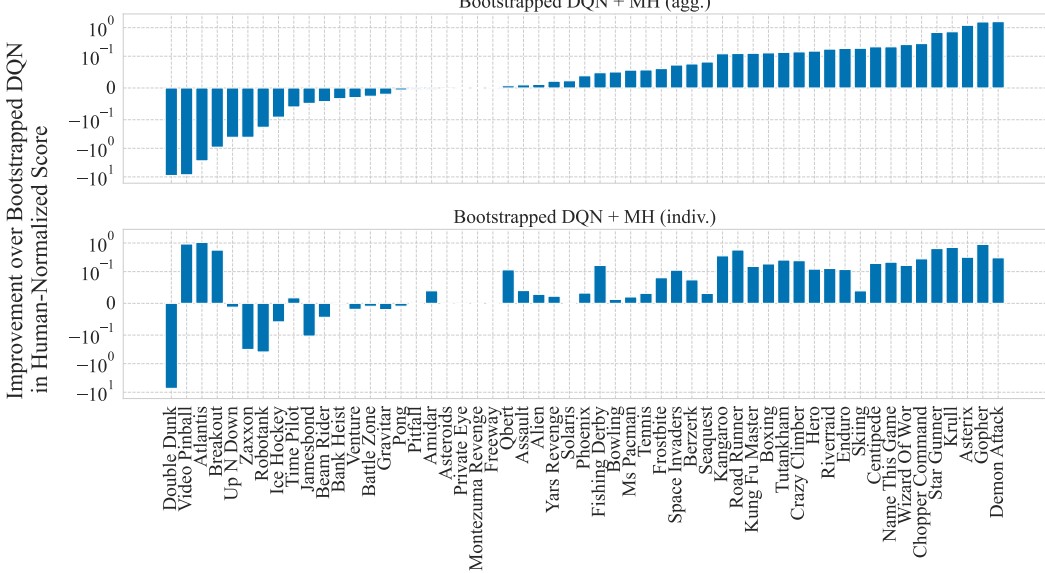

Figure 27: Per-game improvements of Bootstrapped DQN + MH over Bootstrapped DQN in HNS, with and without policy aggregation.

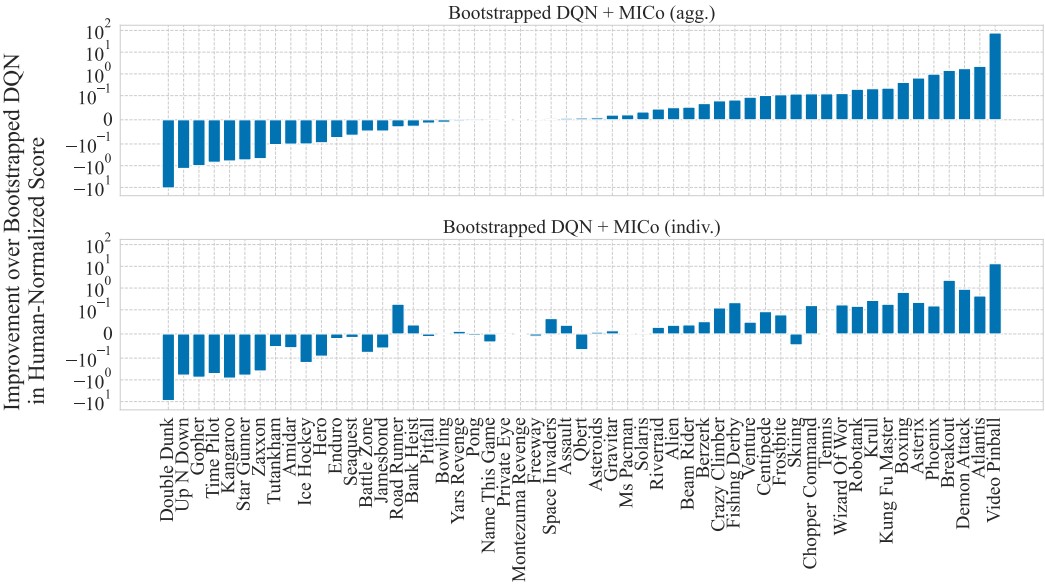

Figure 28: Per-game improvements of Bootstrapped DQN + MICo over Bootstrapped DQN in HNS, with and without policy aggregation.

