# OpenReview forum: "The Curse of Diversity in Ensemble-Based Exploration"
_ICLR.cc/2024/Conference — ICLR 2024 poster_

### Official Review · Reviewer_ZXyY · 2023-10-24

**Soundness:** 4 excellent
**Presentation:** 4 excellent
**Contribution:** 3 good
**Rating:** 8
**Confidence:** 4

**Summary:**

The paper proposes a new way of ensembling RL agents. N neural network value functions are trained, consisting of a convolutional encoder and a value head. Each encoder has a main value head, so we have N main value heads. For each encoder, N value heads are trained with temporal difference, each using as the target one of the main value heads. That gives us N^2 value heads in total. This improves representation learning and data utilization. There are extensive experiment showing that training ensembles of RL agents with other methods are hard, while the proposed method works well. There are experiments on atari with DQN and mujoco with SAC.

**Strengths:**

- The observation that training ensembles of RL agents is hard is interesting and is analyzed well
- The proposed method is novel and sound
- The experiments are comprehensive and demonstrate the benefits of the method

**Weaknesses:**

No major weaknesses

**Questions:**

\-

---

> ### Author Response · Authors · 2023-11-18
> **Author Response**
>
> Thank you very much for your positive comments! We are pleased to hear that there are no major concerns regarding our paper. Your feedback is greatly valued and has been encouraging for us.

---

### Official Review · Reviewer_H7rr · 2023-10-29

**Soundness:** 3 good
**Presentation:** 3 good
**Contribution:** 3 good
**Rating:** 6
**Confidence:** 4

**Summary:**

This work studies the problem of off-policy-ness in ensemble learning. Motivated by the recent discovery in offline RL (i.e., the tandem paper), and empirical observations in ensemble learning methods of BootstrapDQN and naive ensemble SAC, the authors introduce a representation learning approach that considers the off-policy data in auxiliary tasks.
Experiments on continuous control and discrete control demonstrate the effectiveness of the proposed method.


----

After rebuttal, I increased my overall rating from 5 to 6, soundness score from 2 to 3, and contribution score from 2 to 3.

**Strengths:**

I like the way this paper is presented. It is clearly motivated by empirical discoveries, together with reasonings, and followed by solutions to the identified problems.

The authors made great efforts in conducting and presenting experiments. Results are reported in a statistically-identifiable way. I really appreciate it.

**Weaknesses:**

### On high-level Motivation:

I’m lost in the motivation of using ensemble in **policy learning**. As has been demonstrated in [EDAC] and [REDQ], I acknowledge that using ensemble learning for the **value function** could lead to improved performance, as the value can be more accurate, with uncertainty. But what is the motivation for having **multiple policies** for ensemble (because they are sample generators, rather than learners). Should not those samplers aim at more efficiently decreasing the uncertainty in the value function?

This is a concern with the continuous setting. BootstrapDQN is purely a **value-based** method, and in discrete tasks, the epsilon-greedy exploration can be regarded as sampling from an off-policy uniform policy, this should not be a problem.

### On BootstrapDQN:

In BootstrapDQN, different samples are assigned different weights to analog bootstrap sampling. It would not be surprising to see individual BootstrapDQN perform worse than DDQN. To verify the claim that *BootstrapDQN’s performance improvement is purely based on voting* made by the authors, a comparison between Bootstrap Sampling/ normal sampling; and a comparison between majority voting and


Figure 2: The reported results of SAC can not match the open-source implementations. This might be due to the usage of a different size of buffer (200k). Could the authors please explain more about this?


### On Performance:

The performance of CERL seems marginal in Figure 7. Also in Figure 7, BootstrapDQN seems not helpful at all.


### On related work:


This work [https://arxiv.org/pdf/2209.07288.pdf] discussed the ensemble learning that trades off between exploration and exploitation using different reward values may also be worth a discussion.

This work [https://openreview.net/forum?id=NOApNZTiTNU] and [REDQ] discussed the usage of the ensemble in Q-learning. REDQ has been considered to be state-of-the-art since 2021.


References:

[EDAC] An, Gaon, et al. "Uncertainty-based offline reinforcement learning with diversified q-ensemble." Advances in neural information processing systems 34 (2021): 7436-7447.

[REDQ] Chen, Xinyue, et al. "Randomized ensembled double q-learning: Learning fast without a model." arXiv preprint arXiv:2101.05982 (2021).

**Questions:**

Please see the weakness section

---

> ### Author Response · Authors · 2023-11-18
> **Author Response**
>
> Thank you for liking our presentation and providing detailed feedback and suggestions! Please see our response below
>
> ## On high-level motivation
>
> > As has been demonstrated in [EDAC] and [REDQ], I acknowledge that using ensemble learning for the value function could lead to improved performance, as the value can be more accurate, with uncertainty. But what is the motivation for having multiple policies for ensemble
>
> We emphasize that even though both (1) ensemble-based exploration methods such as Bootstrapped DQN and Ensemble SAC and (2) methods such as EDAC and REDQ use ensembles, they use ensembles in completely different ways and purposes. Bootstrapped DQN and Ensemble SAC use ensembles *purely for better exploration*.  In contrast, methods such as EDAC and REDQ use ensembles *purely for improving value estimation*. The concrete differences are detailed below, which should clear up your confusion:
>
> * The number of policies:
>   * Both Bootstrapped DQN and Ensemble SAC maintain $N$ policies. Note that for Bootstrapped DQN the $N$ policies are *implicitly defined by the $N$ value functions*. Most importantly, all $N$ policies are used to collect samples. Concretely, for each episode, one policy is sampled for interaction. **The motivation for having $N$ policies in Ensemble SAC is the same as that of having $N$ different value functions (and hence $N$ *implicit* policies) in Bootstrapped DQN: to explore the state space with a diverse set of policies (explicit or implicit) concurrently in a temporally coherent manner**.
>   * EDAC and REDQ are simply not designed for exploration. They only have one policy and it is the only one used to interact with the environment.
> * How the value functions are used:
>   * In Bootstrapped DQN and Ensemble SAC, *each value function $Q_i$  evaluates one  policies $\pi_i$*. These value functions are *independent* (though they may share networks) in the sense that the TD regression target for each ensemble member is computed independently with their target network. Concretely, the target for $Q_i(s, a)$ is $r(s, a) + \gamma \bar{Q}_i(s’, a_i’)$, where $a_i’\sim\pi_i(a_i’|s’)$. They are not designed to produce more accurate value estimations than single-agent Double DQN or SAC. There is no explicit mechanism that encourages more accurate value estimations such as penalizing Q-value based on uncertainty (though the diverse data may help implicitly).
>   * In EDAC and REDQ, *all the $N$ value functions $Q_{1:N}$ evaluate the same single policy $\pi$*. The $N$ value functions **share a single TD regression target** that is aggregated from all target networks. For examples, in EDAC, the target for $Q_i(s, a)$ -- regardless $i$ -- is $r(s, a) + \gamma\min_j \bar Q_j(s’, a’)$ (ignoring the entropy term), where $a' \sim\pi(a'|s')$. Note how the target values are aggregated with the $\min$ operator to counter overestimation.
>
> In summary, the goal of having multiple policies in Ensemble SAC is to promote better exploration. This is the exactly same motivation as having multiple value functions (which implicitly define multiple policies) in Bootstrapped DQN. In contrast, EDAC and REDQ are not designed for better exploration; there is only one policy in them and it is the only policy interacting with the environment.

---

> > ### Author Response · Authors · 2023-11-18
> > **Author Response (cont.)**
> >
> > ## On BootstrapDQN
> >
> > > In BootstrapDQN, different samples are assigned different weights to analog bootstrap sampling. It would not be surprising to see individual BootstrapDQN perform worse than DDQN. To verify the claim that BootstrapDQN’s performance improvement is purely based on voting made by the authors, a comparison between Bootstrap Sampling/ normal sampling; and a comparison between majority voting and
> >
> > As mentioned in Appendix A, we do *not* use data bootstrapping in
> > Bootstrapped DQN. We just compute minibatch losses as usual without any masking or weighting. Note the original Bootstrapped DQN paper does not find bootstrapping to be useful in Atari games (see Figure 5b in the Bootstrapped DQN paper). As a result, they do not use bootstrapping in their Atari experiment. This is likely the standard way to implement Bootstrapped DQN for Atari.
> >
> > In our revision, we also mentioned this point in Section 2 of the main text to avoid confusion for future readers.
> >
> > For completeness, we also tested Bootstrapped DQN with data bootstrapping (with masking probability $p=0.75$). The results are shown in Figure 20 of Appendix D.3.7. As you suggested, it does damage performance. This is not surprising because masking out samples essentially reduces the number of transitions each member has access to. It also effectively reduces the batch size.
> >
> > We also emphasize that we do not claim “BootstrapDQN’s performance improvement is purely based on voting” (even though our result does suggest that voting is much more important than previously attributed). We have revised Section 3.1 to avoid this misinterpretation.
> >
> > > Figure 2: The reported results of SAC can not match the open-source implementations. This might be due to the usage of a different size of buffer (200k). Could the authors please explain more about this?
> >
> > This is correct. As mentioned in the **Environments and algorithms** paragraph of Section 2 this is done for analysis purposes. Results with larger replay buffers are presented in Figure 4 (right) and Figure 7 (right), which match the performance of open-source implementation, except on HalfCheetah where our implementation slightly underperforms. Note that we use the v4 MuJoCo environments which have some technical differences from the v2 environments the original JAXRL repository used to report results. We also slightly adjusted the implementation of the actor (tanh instead of hard clipping for the `log_std` parameter) for better stability. Therefore, some minor differences in performance are expected.
> >
> > ## On Performance
> >
> > > The performance of CERL seems marginal in Figure 7.
> >
> > Figure 7 only shows aggregate performance over 55 games, which can sometimes be misleading. We invite the reviewer to look at Figure 13 of Appendix D.2.3, which provides a per-game performance summary and should better reflect the benefits of CERL. Also, in MuJoCo tasks, CERL provides significant improvements when the curse of diversity is severe (i.e., when the replay buffer is small), which is exactly what CERL is proposed to fix. On the other hand, we emphasize that our work is the first to identify the curse of diversity phenomenon and CERL is only a first step towards mitigating it.
> >
> > > Also in Figure 7, BootstrapDQN seems not helpful at all.
> >
> > As mentioned above, aggregate performance can be misleading. Please see Figure 2 top right, where we provide a per-game performance summary. In this plot, we see that Bootstrapped DQN behaves very differently from Double DQN, and whether it is helpful is highly environment-dependent.
> >
> > ## On related work
> >
> > Thanks for pointing out these related works and we have included them in the related work section.
> >
> > We hope our response has addressed your concerns. Please let us know if you have any further questions!

---

> > > ### Comment · Reviewer_H7rr · 2023-11-21
> > > **Thank you for the response.**
> > >
> > > Many thanks for the detailed response. My concerns are well addressed, and I have increased my rating accordingly.

---

### Official Review · Reviewer_wgEM · 2023-10-30

**Soundness:** 3 good
**Presentation:** 4 excellent
**Contribution:** 4 excellent
**Rating:** 8
**Confidence:** 5

**Summary:**

The paper introduces a novel observation for ensemble-based exploration: each ensemble member has significantly worse performance than single-agent baselines, even when the aggregate policies outperform them. The authors demonstrate this in experiments with Double DQN and SAC on Atari and Mujoco environments. They show that increasing the replay buffer and sharing layers reduce the effect, but do not lead to better performance. The paper proposes a novel representation learning method (CERL), in which ensemble members predict each other's targets with additional network heads. Results indicate that CERL improves the performance individual ensemble members without loosing too much diversity of their predictions.

**Strengths:**

The reviewer liked the paper a lot. The main hypothesis makes sense and is substantiated in multiple experiments that show the effect nicely. The paper is well written and the figures are clearly readable. More detailed figures for individual environments are provided in the appendix, which is welcome to get an idea how trustworthy the aggregate performance measures are. The proposed method is not terribly innovative, but to the best knowledge of this reviewer novel. The discussion on other representation learning methods is nice, too.

**Weaknesses:**

While the main paper is very well written and the experiments appear quite thorough, the reviewer took issue with the way that some conclusions were presented. In particular the connection to exploration (which is in the title) ignores some major alternative explanations of the results. While the reviewer recommends to accept the paper, some phrases *need* to be changed, and some discussion needs to be added, to prevent the casual reader from misinterpreting the text and results. These are:

1. "First, it challenges some previous work’s claims that the improved performance of approaches such as Bootstrapped DQN in some tasks comes from improved exploration" (p.3): as mentioned earlier in the paper, ensemble methods are widely used to drive exploration into unseen regions of the state-action space. However, the authors claim that "Surprisingly, simply aggregating the learned policies at test-time provides a huge performance boost in many environments" (p.3), and proceed to downplay the effect of exploration in favor of an explanation based on "aggregation". This is a problem in statements like "the two cases[, DQN(indiv.) and Double DQN,] have access to the same amount of data and have the same network capacity" (p.3), as the change in exploration means that the two algorithms are based on different data distribution and will therefore behave differently, irrespective of the on- and off-policy-ness of the data. This potential interaction between exploration (which part of the state-space is covered) and on-policy-ness (which member got to sample it) cannot be distinguished with the presented experiments, but the text reads as if the latter is the only obvious conclusion (see below).

2. "These results confirm our hypothesis regarding the cause of the observed performance degradation" (p.5). This is dangerous abductive reasoning: experimental evidence that is consistent with your hypothesis does not make your hypothesis true (one can only falsify a hypothesis). This may sound like a nitpick, but the paper does not discuss the exploration effect of ensembles enough. For example, in Figure 3 (middle), the BootDQN (indiv.) curve is significantly above the Passive (10%) curve. If indeed Algo 2 and Algo 3 are exactly equivalent except for how often they sample the other (non-passive) ensemble members, and *only* this affects the performance due to off-policy sampling, then both curves (which have seen the same fraction of off-policy samples) should perform identical. The fact that they do not shows that there must be another effect at work here. This could be the exploration, but the experiments do not allow to distinguish these potential effects. To be fair, the reviewer also does not know how to distinguish exploration and on-policy-ness with the given setup, but another setup might be able to (e.g. with intrinsic-reward methods where the exploration can be separated from who acts). The presented analysis is strong enough for publication (a new setup is not needed), but these ambiguities must be discussed!

3. While the reviewer liked most experiments (e.g. Figure 3 and 4 are great!), Figure 5 seems to test DQN with a very small replay buffer, where even the aggregated ensemble does not perform as well as Double DQN. This makes some conclusions suspect. For example, does the aggregated performance improves when more layers are shared? Or does the ensemble become indistinguishable from the baseline? If the authors would have picked a larger replay buffer (e.g. 4M), the performance would improve in the first and worsen in the second case. This would have an impact, as it shows that sharing layers in the ensemble reduces the positive effect it can have (e.g. by exploration).

4. The main hypothesis of the paper is that "the curse of diversity" hurts individual member performance due the large number of off-policy samples. This makes sense, as an individual member can make 90% of all decisions correctly and still have poor performance, whereas a majority vote of the ensemble will agree with high certainty on the correct action. Ultimately this is the reason why BootDQN explores well. However, CERL seems to have almost as much diversity as the L=0 ensemble. CERL is a representation learning algorithm, which seems to be at odds with the main hypothesis. Contrast this with BootDQN which switches ensemble members mid-episode. If this algorithm would improve individual performance, the reviewer would accept it as a fix of the identified problem. With CERL presented as the "solution", the reviewer wonders: is the underlying problem is indeed "diversity"? Please discuss why representation learning fixes the off-policy-ness of the data (there are some attempts, but it is still vague).

In summary: there is a lot to love about the paper, but the authors need to reformulate some critical points (or convince the reviewer that those are not critical). Only under the condition that this happens, this reviewer recommends acceptance.

**Minor comments**

- The figure captions should contain more information about the shown algorithms (ensemble size, replay buffer size, L, ...) to make them more self-contained for cross-reading.
- Ensembles are primarily used because they are good out-of-distribution detectors. An experiment measuring this would have been very welcome.

**Questions:**

- If one accepts the idea that ensembles are primarily for exploration, then the diversity of responses is actually a feature, not a bug. Given that few wrong decisions per episode can significantly reduce the performance of an individual ensemble member, why do the authors actually consider the poor individual performance an issue, in particular when the aggregate performance is better than the baseline (e.g. on Mujoco)?
- How did the baseline (Double DQN and vanilla SAC) explore? If the baseline worked so well, doesn't this imply exploration is not very important in the tested environments? How would the analysis look like if it would concentrate on hard-exploration environments?

---

> ### Author Response · Authors · 2023-11-18
> **Author Response**
>
> Thank you for the insightful comments and detailed feedback! These comments lead us to think more deeply about our results and are very helpful for enhancing the quality of our work. Please see our response below:
>
> ## Response to “Weaknesses”
>
> > **In Weaknesses 1**: This is a problem in statements like "the two cases[, DQN(indiv.) and Double DQN,] have access to the same amount of data and have the same network capacity" (p.3), as the change in exploration means that the two algorithms are based on different data distribution and will therefore behave differently, irrespective of the on- and off-policy-ness of the data. This potential interaction between exploration (which part of the state-space is covered) and on-policy-ness (which member got to sample it) cannot be distinguished with the presented experiments, but the text reads as if the latter is the only obvious conclusion (see below).
>
> > **In Weaknesses 2**: …The fact that they do not show that there must be another effect at work here. This could be the exploration, but the experiments do not allow us to distinguish these potential effects
>
> Thank you for raising the point regarding exploration/coverage. We have included a detailed discussion on this in the “Remark on data coverage” paragraph at the end of Section 3.2. The main points are summarized as follows:
>
> 1. State-action space coverage is *purely a property of the data distribution*, but off-policy-ness depends on *both the data distribution and the policies*. So indeed they are different.
> 2. Our p%-tandem experiment disentangles these two aspects: the active and passive agents are trained on the same data (thus same coverage), but experience different degrees of “off-policy-ness”.
> 3. As a result, our experiment indicates that “off-policy-ness” is detrimental and sufficient to cause the performance degradation we see. At the same time, we recognize that it *does not* imply whether wider state-action space coverage is beneficial (more likely and intuitive) or harmful.
> 4. On the other hand, as you mentioned, the fact that BootDQN (indiv.) performs better than the passive agent indicates that wider coverage is *likely* highly beneficial. Reasoning:
>    1. Due to the presence of $N$ different policies, the individual agents in BootDQN *likely* suffer from more severe off-policy-ness than the passive agent.
>    2. Therefore, if the wider coverage in BootDQN was not beneficial, due to more severe off-policy-ness BootDQN (indiv.) should have performed significantly worse than the passive agent.
> 5. However, unlike the comparison between the active and passive agents which is perfectly controlled, the comparison between BootDQN (indiv.) and the passive agent may have other (unexpected) confounders. Therefore we avoid making conclusive claims about the effect of state-action space coverage in our paper.
>
> We have also added “even though they are trained on different data distributions and thus are expected to behave differently,” before the statement “the agents in these two cases have access to the same amount of data and have the same network capacity” in Section 3.1 to emphasize that the data distributions in these two cases are different.
>
> > **In Weaknesses 1**: the authors claim that "Surprisingly, simply aggregating the learned policies at test-time provides a huge performance boost in many environments" (p.3), and proceed to downplay the effect of exploration in favor of an explanation based on "aggregation"
>
> To avoid misinterpretation, we have revised the second paragraph in Section 3.1 to emphasize that even though we show that majority voting plays a more important role than previously attributed, it does not indicate that exploration/state-action space coverage is not important or beneficial.
>
> > **In Weaknesses 2**: "These results confirm our hypothesis regarding the cause of the observed performance degradation" (p.5). This is dangerous abductive reasoning: experimental evidence that is consistent with your hypothesis does not make your hypothesis true (one can only falsify a hypothesis).
>
> We agree that there could be other factors at play (e.g., state-action space coverage, as discussed), even though we do believe that our results provide strong evidence that off-policy-ness plays a major role in the observed performance degradation. We have changed our statement to “These results offer strong evidence of a connection between the off-policy learning challenges in ensemble-based exploration and the observed performance degradation” as a more accurate conclusion of our results.

---

> > ### Author Response · Authors · 2023-11-18
> > **Author Response (cont.)**
> >
> > > **In Weaknesses 3**: Figure 5 seems to test DQN with a very small replay buffer, where even the aggregated ensemble does not perform as well as Double DQN. This makes some conclusions suspect. For example, does the aggregated performance improve when more layers are shared? Or does the ensemble become indistinguishable from the baseline? If the authors would have picked a larger replay buffer (e.g. 4M), the performance would improve in the first and worsen in the second case.
> >
> > In Figure 5 we use a replay buffer of 1M transitions. This is the “default size” used in most DQN variant papers (e.g., Rainbow, QR-DQN, M-IQN). For analysis purposes, we select the four environments where the curse of diversity is obvious based on preliminary experiments, and that’s why both their `indiv.` and `agg.` performance is worse than Double DQN. For completeness, we have performed the “layer sharing” experiment using a replay buffer size of 4M transitions. The results are shown in Figure 18 of Appendix D.3.5. This does allow BootDQN (agg.) to outperform Double DQN in 3 environments, but the trade-off between the advantages and disadvantages of diversity remains: as we increase the number of shared layers and hence reduce diversity,
> >
> > 1. The curse of diversity, i.e. the gap between Double DQN and BootDQN (indiv.), reduces.
> > 2. The performance gain we get from majority voting, i.e. the gap between BootDQN (agg.) and BootDQN (indiv.), also reduces. An exception is Space Invaders, where the gap seems to slightly increase when L increases from 0 to 3, which might be related to certain properties of this game (that we do not understand at this point).
> >
> > The Nature DQN architecture we use in this work has 5 layers, so BootDQN (L=5) is equivalent to Double DQN. Therefore, further increasing the number of shared layers will not improve performance.
> >
> > Given the above results, we realize that the statement “Most importantly, neither of these techniques allows Bootstrapped DQN (agg.) to outperform Double DQN in a statistically significant way” in our initial submission can be environment- and hyperparameter-dependent. Besides, this statement may obfuscate the main points we are making (i.e., the trade-off between the advantages and disadvantages of diversity) with Figure 5, so we have removed it in our revision.
> >
> > > **In Weaknesses 4**: However, CERL seems to have almost as much diversity as the L=0 ensemble. CERL is a representation learning algorithm, which seems to be at odds with the main hypothesis… With CERL presented as the "solution", the reviewer wonders: is the underlying problem is indeed "diversity"? Please discuss why representation learning fixes the off-policy-ness of the data (there are some attempts, but it is still vague).
> >
> > As mentioned at the beginning of Section 3, two main factors contribute to the curse of diversity: (1) the off-policy-ness of the data w.r.t the agents (which largely depends on ensemble diversity) and (2) the agent’s inefficiency to utilize such off-policy data. **CERL does not fix (1) but tries to fix (2)**. It is easier to understand this point by realizing that a major part of the difficulty of off-policy learning in *deep* RL is about generalization: how well an agent can learn from off-policy data depends a lot on how well the Q network can *generalize* to state-action pairs that have high probability under the current policy but are *under-represented* in the data. This is often studied under the name of “extrapolation error” in off-policy/offline RL [1, 2]. Representation learning can potentially improve the network’s generalization abilities and thus can improve the agent’s capability to learn from off-policy data.
> >
> > An example of “proper representations” helping off-policy learning is given in the tandem RL paper [1]. They show that if the passive agent uses the active agent’s representation (by sharing the first few layers but not allowing the passive agent to update them), the tandem effect can largely be mitigated. In the context of ensemble RL, the equivalent of this approach is just sharing layers of the Q-networks, which we have shown can compromise diversity and the advantages of ensembling (e.g. exploration and majority voting). The motivation of CERL is just to achieve the similar representation learning effect of network sharing (i.e., learning multiple value functions at the same time) without actually needing to share the networks
> >
> > We have included a sentence in the first paragraph of Section 4 to further clarify the motivation of CERL.

---

> ### Author Response · Authors · 2023-11-18
> **Author Response (cont.)**
>
> > **In Weaknesses 4**: Contrast this with BootDQN which switches ensemble members mid-episode. If this algorithm would improve individual performance, the reviewer would accept it as a fix of the identified problem.
>
>  We have tested this variant In Figure 17 of Appendix D.3.4 . Specificially, we switch ensemble members at each step instead of only at the start of an episode. As shown in the result it does not mitigate the curse of diversity. This is not surprising because it does not change the proportion of “self-generated” data for each agent in the replay buffer.
>
>
> ## Response to “Minor comments”
>
> >  The figure captions should contain more information about the shown algorithms (ensemble size, replay buffer size, L, ...) to make them more self-contained for cross-reading.
>
> Thanks for your suggestion. We have included these in the captions in our revision.
>
> > Ensembles are primarily used because they are good out-of-distribution detectors. An experiment measuring this would have been very welcome.
>
> Thank you for your suggestion regarding OOD detection. We agree it's an important aspect. However, we intentionally focused on the off-policy learning aspects of ensemble-based exploration for this paper, and incorporating OOD detection might go beyond our current scope. We look forward to exploring this in future research.
>
> ## Response to “Questions”
>
> > If one accepts the idea that ensembles are primarily for exploration, then the diversity of responses is actually a feature, not a bug. Given that few wrong decisions per episode can significantly reduce the performance of an individual ensemble member, why do the authors actually consider the poor individual performance an issue, in particular when the aggregate performance is better than the baseline (e.g. on Mujoco)?
>
> It is widely accepted that a diverse ensemble can potentially result in better exploration and a more robust aggregate policy. The point of our work is not to deny these advantages, but to point out that despite these benefits, diversity also has a negative side effect. **The fact that its advantages (e.g., majority voting) sometimes outweigh its disadvantages (off-policy-ness) does not mean that the disadvantages do not exist or are not an issue.** This has practical significance. For example, without realizing the negative effect of diversity, one might think that having more ensemble members is always better (if we ignore computational costs), which is not the case as we have shown in Figure 5. The cause of this non-monotonic behavior is obvious from the individual agents’ performance but not from the aggregate policy’s performance. Also, knowing the cause of the curse of diversity allows targeted solutions that may give rise to better algorithms. CERL is just a first step in this direction. Finally, as mentioned in the second paragraph of Section 3.1, there are some scenarios such as hyperparameter sweep with ensembles where we do care about the performance of the individual ensemble members.
>
> > How did the baseline (Double DQN and vanilla SAC) explore? If the baseline worked so well, doesn't this imply exploration is not very important in the tested environments? How would the analysis look like if it would concentrate on hard-exploration environments?
>
> Double DQN uses epsilon-greedy and vanilla SAC just uses stochastic policy + entropy regularization. These are very simple baselines and the performance on the tested environment is far from saturated, so the results by no means imply exploration is not very important in the tested environments. Note that the suite of 55 Atari games we tested contains several games that are traditionally considered hard exploration problems (e.g., Montezuma’s Revenge. For a complete list see Table 1 in [3]). As shown in Figure 2 (top right) and Figure 10 of our paper, Bootstrapped DQN does not show clear advantages over Double DQN in most of these games. In fact, in the original Bootstrapped DQN paper, the author also finds that “On some challenging Atari games where deep exploration is conjectured to be important our results are not entirely successful, but still promising” and shows that the improvements could be highly unstable (e.g., on Montezuma’s revenge). This is not surprising as these hard exploration games often require special events that are very unlikely with primitive exploration techniques such as epsilon-greedy or ensemble.
>
> We hope our response has addressed your concerns. Please let us know if you have any further questions!
>
> ## References
>
> [1] Ostrovski, Georg et al. “The Difficulty of Passive Learning in Deep Reinforcement Learning.” Neural Information Processing Systems (2021).
>
> [2] Fujimoto, Scott et al. “Off-Policy Deep Reinforcement Learning without Exploration.” International Conference on Machine Learning (2018).
>
> [3] Bellemare, Marc G. et al. “Unifying Count-Based Exploration and Intrinsic Motivation.” Neural Information Processing Systems (2016).

---

> > ### Comment · Reviewer_wgEM · 2023-11-19
> > **I vote for accept**
> >
> > Thanks to the authors for their exhaustive answer and the changes to the paper. In particular Figure 18 shows exactly what I expected: increasing the shared layers removes the off-policy-ness of the data (because the ensemble members become indistinguishable), but also loose the beneficial effect of the ensemble, in all likelihood exploration. The other changes are also welcome and address most of my criticisms.
> >
> > In conclusion, I vote to accept the paper in its current state.

---

### Official Review · Reviewer_5Qz1 · 2023-11-02

**Soundness:** 3 good
**Presentation:** 4 excellent
**Contribution:** 3 good
**Rating:** 8
**Confidence:** 4

**Summary:**

This paper is a primarily empirical paper investigating what the authors term "the curse of diversity". This phenomenon is where
individual members of a data-sharing ensemble underperform relative to their single agent counterparts due the high proportion of off-policy data relative to each individual ensemble member and the agents' inability to learn effectively from such off-policy data.

The authors first demonstrate in the bootstrapped DQN setting (a common ensemble Q-learning method) that individual ensemble members perform poorly. Then, the authors run experiments to show the underperformance of an individual learner when there is a low self-generated to "other-generated" ratio in the replay buffer. This experiment, along with some reasoning, leads to the conclusion that the curse of diversity indeed exists.

They then investigate whether the curse of diversity can be ameliorated through larger replay buffers. The results are mixed, suggesting larger replay buffers may sometimes mitigate the curse, though largely do not. They then investigate ways to reduce the curse of diversity by directly reducing diversity in the ensemble, but find that it can adversely impact the underlying ensemble method.

They then introduce Cross-Ensemble Representation Learning (CERL), as a mechanism to mitigate the curse while preserving the benefits of the underlying ensemble method. CERL works by adding auxiliary tasks to ensemble members, having them learn value functions of other ensemble members and find that CERL mitigates the curse of diversity without adversely impacting performance.

**Strengths:**

Presentation - The paper is extremely clear and excellently written. The problem, motivation, and experiments are articulated very clearly. I like that they look introspectively at their own experiments and reason clearly about what can be inferred/concluded from their experiments without making unreasonable intellectual leaps.

Contributions:
1. They show that perhaps the aggregation/majority voting aspect of ensembling methods may contribute to improved performance more than previously attributed. Previously, the intuition/hypothesis was more along the lines of ensembling methods inducing diversity in the data, which leads to higher-quality value estimates.
2. They demonstrate that the curse of diversity exists, at least for ensemble Double DQN methods.
3. They look at some natural ways to mitigate the curse of diversity and analyze their effectiveness.
3. They start the steps towards mitigating the curse of diversity

Experimentation: The experiments are reasonably thought out, and cover a wide range of environments (Atari/Mujoco).

**Weaknesses:**

I do not have any many major qualms with the paper, but I'll list a few thoughts.

Perhaps this is out of scope of the paper, but I do feel it's difficult to draw conclusions about deep RL more broadly without investigating distributional RL. For example, this paper (https://openreview.net/pdf?id=ryeUg0VFwr) shows that distributional RL will likely do better with this off-policy data. It would be interesting to investigate the extent of this phenomena in the distributional setting.

CERL does seem to scale poorly with the ensemble size, since if the ensemble size is N, there are N x N Q-functions. In practice, this does not seem like a major problem.

For the bar plots in Figure 2, it might be helpful to plot the top plot minus the bottom plot to highlight the point.

**Questions:**

Question: Did you try different levels of passivity? Other than 10%?

---

> ### Author Response · Authors · 2023-11-18
> **Author Response**
>
> Thank you for appreciating our contributions and providing valuable feedback! Please see our response below:
>
> > Perhaps this is out of scope of the paper, but I do feel it's difficult to draw conclusions about deep RL more broadly without investigating distributional RL
>
> This is a great suggestion and we have added experiments on four Atari games with QR-DQN. The results are presented in Figure 16 of Appendix D.3.3. The significant performance gap between QR-DQN and Bootstrapped QR-DQN (indiv.) shows that the curse of diversity is also present in distributional RL. Note that due to limited time, we are not able to finish the full 200M frames of training. We will include the full results when the experiments are finished.
>
> > CERL does seem to scale poorly with the ensemble size, since if the ensemble size is N, there are N x N Q-functions. In practice, this does not seem like a major problem.
>
> In practice, CERL scales very well with ensemble size. As mentioned in the “Method” paragraph in Section 4, to implement CERL we only need to increase the output dimension of the *last linear layer* of the network (note this is the same architectural modification involved in distributional RL). So even though CERL requires N x N Q-functions as opposed to N Q-functions in a standard ensemble, the increase in the number of parameters and wall clock time is very small. For example, adding CERL to Bootstrapped DQN (L=0, N=10) increases the number of parameters by no more than 5% and the increase in wall clock time is barely noticeable. Even if we use N=20 which is already much larger than commonly used in the literature, applying CERL increases the number of parameters by no more than 11% and the increase in wall clock time is less than 5%.
>
> > For the bar plots in Figure 2, it might be helpful to plot the top plot minus the bottom plot to highlight the point.
>
> Thanks for your suggestion! We have included this figure in Figure 12 of Appendix D.2.2. We do not include it in the main text to save space for other changes, and also because this plot mainly shows the benefits of majority voting which is not the primary focus of our work.
>
> > Question: Did you try different levels of passivity? Other than 10%?
>
> We have also tried 0%, 5%, and 20% in four Atari games. The results are shown in Figure 19 of Appendix D.3.6. As expected, as the degree of passivity reduces (i.e., larger $p$), the active/passive agents’ performance gap also reduces.
>
> Please let us know if you have any further questions or suggestions!

---

> > ### Comment · Reviewer_5Qz1 · 2023-11-19
> > **Reviewer Response to Author Response**
> >
> > Thank you for your detailed response and for addressing my comments.

---

### Author Response · Authors · 2023-11-18
**General Response**

We thank all the reviewers for their time and effort in reviewing our paper and providing valuable feedback. We have addressed their concerns individually in separate responses. Here we summarize the main changes we made to our paper. **The major changes are highlighted in blue in our revision**.

- As suggested by reviewer 5Qz1,
  - We included an experiment where we use QR-DQN as the base algorithm in Bootstrapped DQN in Figure 16 of Appendix D.3.3. This shows the curse of diversity is also present in distributional RL.
  - We visualized per-game performance improvements due to majority voting in Figure 12 of Appendix D.2.2
  - We performed the p%-tandem experiments with different levels of passivity (other than $10\%$) in 4 games in Figure 19 of Appendix D.3.6.
- As suggested by reviewer wgEM,
  - We revised the text of Section 3.1 to emphasize our result does not indicate that exploration is not beneficial or important. We also included a detailed discussion on the potential role of exploration/state-action space coverage at the end of Section 3.2.
  - We repeated the layer-sharing experiment in Figure 5 (right) with a larger replay buffer of size 4M. The results are shown in Figure 18 of Appendix D.3.5.
  - We tested a variant of Bootstrapped DQN where we switch ensemble members mid-episode (i.e., at each step) in Figure 17 of Appendix D.3.4. This variant does not mitigate the curse of diversity.
- As suggested by reviewer H7rr,
  - In the paper, following the original Bootstrapped DQN paper’s setup for Atari, we *do not* use data bootstrapping. For completeness, we performed an experiment where we *do* use data bootstrapping in Figure 20 of Appendix D.3.7. This damages performance.
  - We expanded the related work in Section 5. In particular, we included ensemble RL methods for purposes other than exploration.
- Other changes:
  - We fixed a plotting error in Figure 13 of Appendix D.2.3 in the revision (originally Figure 15 of Appendix D.4.1 in the initial submission) that caused the per-game improvements of Bootstrapped (indiv.) + CERL over Bootstrapped (indiv.) to be incorrect. The result remains qualitatively the same.
  - We moved the preliminary investigation of other representation learning methods (originally at the end of Section 4) to Appendix D.4.5 to save space for other more important changes to the main text. We also expanded the results from 4 games to all 55 games and fixed a bug in the results of Double DQN + MH that caused it to behave the same as Double DQN.

---

### Meta-Review · Area_Chair_bSV2 · 2023-12-05

**Metareview:**

The reviewers were excited about this paper and gave it good scores, I recommend accepting it.

**Justification For Why Not Higher Score:**

The findings in the paper are interesting but quite expected: diverse policies make of policy learning to challenge. The method of using auxiliary losses has been tried before and the results only demonstrate a small improvmeent over the baseline.

**Justification For Why Not Lower Score:**

There are enough interesting results in this paper to meet the ICLR bar.

---

### Decision · Program_Chairs · 2024-01-16

Accept (poster)